

# A physical protocol for observers near the boundary to obtain bulk information in quantum gravity

**Chandramouli Chowdhury[1]⋆, Olga Papadoulaki[2]† and Suvrat Raju[1]‡**

**1** International Centre for Theoretical Sciences, Tata Institute of Fundamental Research, Shivakote, Bengaluru 560089, India.
**2** International Centre for Theoretical Physics, Strada Costiera 11, 34151 Trieste, Italy.

⋆ chandramouli.chowdhury@icts.res.in, † papadoulaki@ictp.it, ‡ suvrat@icts.res.in

## Abstract

We consider a set of observers who live near the boundary of global AdS, and are allowed to act only with simple low-energy unitaries and make measurements in a small interval of time. The observers are not allowed to leave the near-boundary region. We describe a physical protocol that nevertheless allows these observers to obtain detailed information about the bulk state. This protocol utilizes the leading gravitational back-reaction of a bulk excitation on the metric, and also relies on the entanglement-structure of the vacuum. For low-energy states, we show how the near-boundary observers can use this protocol to completely identify the bulk state. We explain why the protocol fails completely in theories without gravity, including non-gravitational gauge theories. This provides perturbative evidence for the claim that one of the signatures of holography — the fact that information about the bulk is also available near the boundary — is already visible in the low-energy theory of gravity.



# 1 Introduction

A recent paper [1], which extended previous ideas [2–5], argued that a careful consideration of low-energy quantum gravity already suggests that information about the interior of a spacetime can be obtained from measurements near its boundary. This result holds both in asymptotically AdS spacetimes and in asymptotically flat spacetimes. In this paper, we present a concrete physical protocol that allows observers near the boundary of global AdS to extract information about the bulk, when the system is in a low-energy state.

Consider a set of observers who live near the boundary of global AdS. The observers are spread out all over the sphere near the boundary. However, they are equipped with detectors that function only in the time-band $[0, \epsilon]$, and so they cannot make any measurements beyond this time-interval. The geometry of our setup is shown in Figure 1a. Restricting the observers to a time-band near the boundary provides a precise version of a physically-similar picture, shown in Figure 1b, where one allows the observers to explore the spacetime on a single time-slice but not enter the bulk region for $r < \cot(\frac{\epsilon}{2})$. We explain the setup in more detail in section 2.

The observers live in a pure state of the theory that is well-described by an excitation of quantum fields above the global AdS vacuum. We would like to understand how much information such observers can glean about this excited bulk state. We will follow the textbook framework used in discussions of quantum information. We assume that the observers have access to a number of identically prepared systems. The observers can manipulate each system by acting with local unitary operators, and make measurements of Hermitian operators in their region of spacetime. They can also classically communicate the results of their measurements to one-another.

In a local quantum field theory, it is clear that the observers cannot determine much about the bulk state since a large part of the spacetime is just inaccessible to them. It is commonly believed that leading gravitational effects should not modify this conclusion significantly, i.e., a common belief is that in the presence of weak gravity, unless the observers do something that is extraordinarily complicated, they should still not be able to learn much about the bulk state.

But, in this paper, we would like to present a surprising conclusion. When leading gravita-

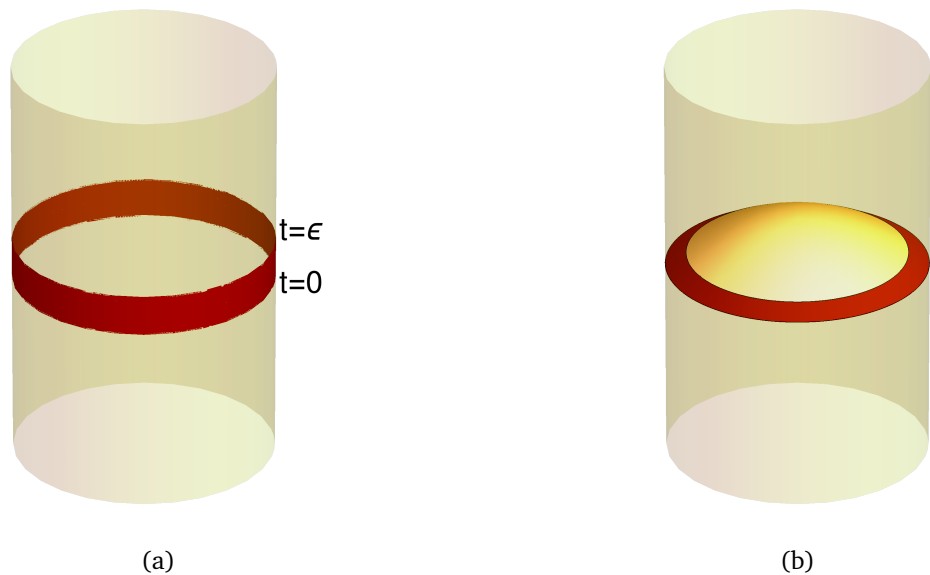

(a)                                                                    (b)

Figure 1: *On the left, we have the precise setup of this paper. The observers can make manipulations and measurements in a small time band (shaded in red) near the boundary of AdS. This is physically similar to the picture on the right, where the observers can explore an annular region near the boundary of a Cauchy slice that runs through the bulk.*

tional interactions are switched on, even if the observers are allowed to act only with simple unitaries, and make only simple measurements, it is possible to develop a protocol that allows the observers to completely determine the state of bulk quantum fields.

For the impatient reader, we immediately explain the main technical idea. First, consider the simplest example. Say that the observers are given the task of verifying whether the state in the bulk — which we denote by $|g\rangle$ and normalize using $\langle g|g\rangle = 1$ — is the global AdS vacuum, which we denote by $|0\rangle$, or some other state. In a local quantum field theory, this task is impossible: in a local QFT, the states $|0\rangle$ and $U_{\mathrm{bulk}}|0\rangle$, where $U_{\mathrm{bulk}}$ is a unitary operator localized near the middle of AdS on some time slice, are completely indistinguishable if one is restricted to observations near the boundary on the same time slice. But, in the presence of gravity, the observers can use gravitational effects to measure the Hamiltonian $H$, which is just given by integrating a particular component of the metric on the boundary sphere. In global AdS, the possible answers for $H$ are quantized. We assume that the observers can measure $H$ with sufficient accuracy to distinguish the first excited state in global AdS from the vacuum. Since $H$ is an operator in the quantum theory, by the standard rules of quantum mechanics, the observers can get different possible results for their measurement. For this task, all they need to do is to determine the *relative frequency* with which their measurement yields 0. By the Born rule, this is given by $\langle g|P_0|g\rangle$ where $P_0 = |0\rangle\langle 0|$ is the projector onto the vacuum. If this is 1, they know the global state is the vacuum, and otherwise it is not.

The skeptic might feel that this is a special case since the vacuum is uniquely identified by a conserved charge. So we now give the observers a harder task. Let $X$ be some simple Hermitian operator accessible to the observers, and consider the state $|X\rangle = X|0\rangle$. The observers are now given the task of determining if the global state $|g\rangle$ coincides with $|X\rangle$. Once again, in a local quantum field theory, this task is impossible. The observers cannot distinguish between $|X\rangle$ and $U_{\mathrm{bulk}}|X\rangle$. But this task can be accomplished using gravitational effects.

For simplicity, let us assume that the observers determine, using the procedure above, that $\langle 0|g\rangle = 0$. Then they can complete their task using a two step process.

1. First, they act on the state with a one-parameter family of *unitary* operators, $U = e^{iJX}$, for various values of $J$, near $J = 0$.

2. After this manipulation, they measure the energy, and determine the relative-frequency with which they obtain 0, to *second order in $J$*.

A very simple calculation yields the expected answer. Expanding the unitary operator to second order in $J$,

$$\langle g|e^{-iJX}P_0 e^{iJX}|g\rangle = \langle g|(1 - iJX - \frac{1}{2}J^2 X^2)P_0(1 + iJX - \frac{1}{2}J^2 X^2)|g\rangle + \mathrm{O}\left(J^3\right). \quad (1)$$

Using the fact that $P_0 = |0\rangle\langle 0|$ and that $\langle 0|g\rangle = 0$ and also that $X|0\rangle = |X\rangle$ we see that

$$\langle g|e^{-iJX}P_0 e^{iJX}|g\rangle = J^2|\langle g|X\rangle|^2 + \mathrm{O}\left(J^3\right). \quad (2)$$

So the protocol used by the observers directly yields the magnitude of the overlap of $|g\rangle$ with $|X\rangle$. If this is $\langle X|X\rangle^{\frac{1}{2}}$, the observers know the two states are the same, and they differ otherwise.

The protocol that we outline in section 3 is just a souped-up version of the two examples that we provided above. Due to the entanglement present in the vacuum, it turns out that states of the form $X|0\rangle$, where $X$ is a low-energy Hermitian operator, near the boundary generate a *basis* for the entire low-energy Hilbert space. By repeating the procedure above for various choices of the operator $X$, we show that the observers can unambiguously identify the bulk state. The technical subtlety is simply that the Hilbert space is a complex vector space, whereas Hermitian operators form a real vector space; so, we need to do work to extract some phases from within the state. We also address other special cases, including the case where $\langle g|0\rangle \neq 0$ in section 3.

We emphasize that even in a local quantum field theory, operators of the form $X|0\rangle$ form a basis. But this property of the vacuum (called cyclicity) is of absolutely no use to near-boundary observers in their task of gleaning bulk information without the operator $P_0$, which they can access only in a theory of gravity. In fact, in section 4, we explain that without gravity, the near-boundary observers obtain exactly *zero* information about the state near the center of AdS. Even in a non-gravitational gauge theory, the so-called "split property" tells us that it is possible to arrange matters so that every observation made by the near-boundary observers yields the same result that it would have yielded in the vacuum state, even if the state near the center of the AdS is completely different. Likewise, if one retains gravity but takes the classical limit, the observers are again unable to discern details of the bulk state. (See section 3.3 of [1] and also the discussion in [6].)

In this paper, we do not explore the implications of our analysis for black holes. But it appears to us that the effect above is closely tied to a key conclusion that has emerged from studies of black hole evaporation: the idea that degrees of freedom in the interior of the black-hole have a dual description in the exterior. In the context of discussions of the black hole information paradox a few years ago [7, 8], this point was emphasized in [9] and in parallel work [10, 11]. Subsequently, a similar point was also made in the ER=EPR proposal [12]. When one considers black holes in asymptotically flat space, an extrapolation of these arguments suggests [1] that information about the microstate can always be extracted by measurements outside the black hole. On the other hand, a different setup has also been studied in a number of recent papers [13–17] which consider black holes in AdS that evaporate into baths without dynamical gravity. Even here, precisely in line with previous expectations, the important physical point is that at late times the black-hole interior is described by degrees of freedom in the bath. Neglecting this identification leads to paradoxes, not only about black holes but even about empty AdS [18]. The analysis in this paper and in [2] is relevant to this story because it provides a clear *physical origin* of these effects that identify degrees of freedom

in one region with those in another region, in a Lorentzian setting, and without invoking any indirect arguments.

The results in this paper also imply that when standard quantum information measures are applied to the geometry shown in Figure 1b, then the answers obtained after including the effects of dynamical gravity are very different from the answers without dynamical gravity. In particular, since information about the entire Cauchy slice is already present in the dark-red "annular region" of Figure 1b, the von Neumann entropy of that region is zero when the global state is pure. Similarly if one considers two different states, then the relative entropy of the states with respect to the "algebra" of the annular region is the same as the relative entropy with respect to the "algebra" on the full Cauchy slice.

In [19], it was suggested that usual notions of quantum information-localization could be recovered by considering only "simple operators". This was also the motivation behind the introduction of the "little Hilbert space" [20] or "code-subspace" [21]. But the surprising aspect of the analysis here is that it is performed *entirely* within the code subspace. Note that these comments are not in contradiction with the results of [22, 23] since the annular region of Figure 1b is not the entanglement wedge of any region on the boundary. We discuss this issue, and also some possible caveats in section 5.

The analysis presented here can be thought of as a perturbative check of the idea that holography [24–26] is implicit in gravity, even from a low-energy perspective. We focus on AdS to avoid some of infrared intricacies of flat space. But we will return to a consideration of flat space in forthcoming work.

An outline of our paper is as follows. We frame the problem precisely in section 2. Section 3 contains the central part of our analysis, and we explain how bulk information can be extracted from manipulations and measurements near the boundary. We explain in section 4 why this protocol fails in theories without gravity, and conclude with a discussion of some implications and subtleties in section 5. We are frequently asked whether our protocol would work in the presence of global symmetries, and so we include a special example showing how to deal with global symmetries in section 3.

## 2 Setup

In this section, we clearly outline our physical setting, the task that the near-boundary observers are given, and the precise powers that they have.

We will consider a spacetime that, asymptotically, tends to global $AdS_{d+1}$.

$$ds^2 \underset{r\to\infty}{\longrightarrow} -(1+r^2)dt^2 + \frac{dr^2}{1+r^2} + r^2 d\Omega_{d-1}^2. \tag{3}$$

In our setup, gravity is weak, but it is dynamical. The AdS radius sets a natural energy scale and, in these units, the scale at which gravitational effects become strong is denoted by $N = G^{\frac{-1}{(d-1)}}$, where $G$ is the low-energy Newton's constant. The physical assumption in our analysis is that the low-energy predictions obtained by straightforwardly quantizing gravity are not affected by UV-effects.

There may be additional dynamical fields in the theory, including string-theoretic excitations. We will assume for simplicity, as is standard in AdS/CFT discussions, that there is no hierarchy of interactions and all tree-level interactions are controlled by the parameter $\frac{1}{N}$, but this assumption can be relaxed as explained in Appendix A. The detailed field-content of the theory will not be important and, apart from the graviton, we will use massive scalars below to discuss the effect of dynamical fields. If $\phi$ is such a field of mass $m$, then we will consider

normalizable excitations of this field with a boundary value

$$\phi(t,r,\Omega) \xrightarrow[r\to\infty]{} \frac{1}{r^\Delta} O(t,\Omega), \tag{4}$$

where $\Delta = \frac{d}{2} + \sqrt{(\frac{d}{2})^2 + m^2}$. The operators $O(t,\Omega)$ restricted to the time-interval $t \in [0,\epsilon]$ are the natural observables in this setup.

The low-energy Hilbert space can be obtained by quantizing the dynamical fields. A standard analysis tells us that this Hilbert space contains a unique vacuum, $|0\rangle$, that is separated from the lowest excited state by a gap proportional to the AdS scale.

The global state that the observers are meant to probe is denoted by $|g\rangle$ throughout the paper. This is a pure state. It is also a low-energy state in the following sense. We introduce a UV-scale $\Lambda$. This scale *defines* what we mean by "low energy" whenever we use the term below. $\Lambda$ is a user-defined scale with the property that it is parametrically smaller than the Planck scale: $\Lambda \ll N$. Then we demand that

$$1 - \|P_{E<\Lambda}|g\rangle\|^2 \ll 1, \tag{5}$$

where $P_{E<\Lambda}$ is the projector onto states with energy lower than $\Lambda$. Note that the condition above allows the the state to have some small components of high-energy. Such high-energy tails are invariably generated by the action of local unitaries. But these tails will not be of interest to us.

## 2.1 Task of the observers

The observers have a simple task. They need to determine the global state $|g\rangle$ up to a given accuracy. More precisely, the observers are challenged with identifying a state $|g_{\text{est}}\rangle$ such that

$$1 - |\langle g_{\text{est}}|g\rangle|^2 \le \delta, \tag{6}$$

where $\delta$ is a parameter with the property that $\delta \ll 1$ but $\delta \gg \frac{1}{N}$. This means that we require the observers to determine the state to a high-level of accuracy, but not such a high-level that the accuracy competes with the ratio of the cosmological scale to the Planck scale.

Of course, if the observers are allowed to directly probe the bulk, this is a straightforward task. But, as we describe below, the observers are restricted to a near-boundary region and it is only by using uniquely gravitational effects that they will be able to perform this task.

## 2.2 Abilities and limitations of the observers

The observers are given access to a number of identically prepared systems, all in the state $|g\rangle$. They are allowed to conduct multiple experiments and collate the results of these experiments in order to identify the close-enough state $|g_{\text{est}}\rangle$. One way to envision the setup is to think of a larger spacetime, in which local patches are well approximated by global AdS. These local patches are then prepared in identical states. The observers make measurements in each local patch, and then travel across the larger spacetime to collate the measurements from different patches.

We emphasize that the need for identical copies is not special to our protocol but is a very basic requirement in any quantum-information analysis. Since measurements are probabilistic, a single copy of the system cannot be used to determine its state. In particular, even bulk correlation functions, which are expectation values of products of operators, can only be measured by averaging the results of measurements in identical systems. Therefore even if the observers were to explore the bulk to obtain information, and not use our protocol at all, they would still require multiple identical copies in order to be able to fix the bulk state.

The interesting restrictions arise in the sort of manipulations and measurements that the observers are allowed to make, and we describe these in turn below.

### 2.2.1 Allowed manipulations

In quantum mechanics, it is standard to allow observers to manipulate the system by acting with unitary operators. Note that it is *not* permissible to "act" on a state with arbitrary Hermitian operators. But Hermitian operators can be added to the Hamiltonian of the theory, and this results in a unitary transformation of the state.

We want to remain within the realm of the low-energy theory. We will do this by allowing the observers to *modify* the state through only low-energy simple unitaries. The allowed unitaries depend on a parameter $J$ and we demand that

$$U = 1 + iJX + \mathrm{O}\left(J^2\right), \tag{7}$$

where $X$ is a low-energy, Hermitian operator from the time band $[0, \epsilon]$ near the boundary of AdS. The operator $X$ must also be a simple operator which means that when it is expressed as a polynomial in the elementary observables $O(t, \Omega)$ of (4) it does not involve any terms of degree higher than $\Lambda$. We will only consider this unitary in the vicinity of $J = 0$. In (7) the reason that we have not written the $\mathrm{O}\left(J^2\right)$ term explicitly is that, as we show below, it drops out of the analysis.

The response of the system under the action of the unitary above can be written as a *modification of the state*, $|g\rangle \to |g_{\mathrm{mod}}\rangle$ where

$$|g_{\mathrm{mod}}\rangle = |g\rangle + iJX|g\rangle + \mathrm{O}\left(J^2\right). \tag{8}$$

We pause to mention two points.

1. There are several physical ways of generating the unitary action (7), and our protocol is insensitive to the method used. In the introduction, to keep the notation simple, we simply used the unitary $e^{iJX}$. But, physically it may be more natural for the observers to turn on a source near the boundary that deforms the Hamiltonian in the time-interval $[0, \epsilon]$ by a term, $-Jx(t)$, where $\int_0^\epsilon x(t)dt = X$. The precise effect of this deformation is an action by the unitary $\mathcal{T}\left\{e^{i\int Jx(t)dt}\right\}$ where $\mathcal{T}$ is the time-ordering symbol. But to first order this unitary also coincides with (7), which is all that we require.

2. We will consider many different manipulations of the state below. But, in order to avoid introducing a plethora of symbols, the relevant unitary is always denoted by $U$ and the the modified state is always represented by $|g_{\mathrm{mod}}\rangle$. So, the notation $|g_{\mathrm{mod}}\rangle$ may refer to different states below, but in each case it will appear immediately after we explain the manipulation that produces it.

### 2.2.2 Allowed measurements

The observers are allowed to perform measurements of low-energy operators near the boundary that are localized in the time-band $[0, \epsilon]$. We are particularly interested in a measurement of the *energy* through the metric. In a gravitational theory in AdS, when Fefferman Graham gauge is chosen near the boundary, the energy of the state is given by the subleading falloff of the metric [27].

$$H = \frac{d}{16\pi G} \lim_{r\to\infty} r^{d-2} \int h_{00}(r, t, \Omega)d^{d-1}\Omega, \tag{9}$$

where $h_{\mu\nu}$ is the deviation of the metric from the global AdS metric displayed in (3). This is just a manifestation of the Gauss law. (A gauge-invariant expression for the energy can be found in [28].) We would like to make a few comments.

1. Since the energy is a delocalized observable, it can be measured in two ways. First, we may consider a team of observers spread out at very large $r$ and all points of the sphere. Each of these observers measures the local value of the metric, and the observers then add their results to obtain the expression for (9). Alternately, a single observer may use multiple identically prepared systems, make measurements at different points on the sphere in different systems, and then add up her results.

2. The energy is a *quantum mechanical* observable. This means that, except in energy eigenstates, its measurement does *not* yield a definite value. However, in global AdS, the possible values obtained upon measuring the energy are *quantized* since energy eigenstates are separated by a gap that is proportional to the inverse AdS length.[1] Both the quantum fluctuations, and the quantized nature of the energy will be useful for us.

3. When the observers measure the energy, there is some probability that they might obtain the answer 0. By the standard rules of quantum mechanics, in the state $|g_{\text{mod}}\rangle$, this probability is given by

$$\langle g_{\text{mod}}|P_0|g_{\text{mod}}\rangle, \tag{10}$$

where $P_0 = |0\rangle\langle 0|$. We will be interested in this probability to obtain 0, not only in the original state, but also after we have manipulated the state.

4. We *do not* require arbitrary accuracy in the measurements above. For instance, as in any quantum mechanical system, to truly measure (10) to arbitrary accuracy would require an infinite number of systems. Here we will be satisfied if the observers can make measurements so that

$$|\langle g_{\text{mod}}|P_0|g_{\text{mod}}\rangle_{\text{measured}} - \langle g_{\text{mod}}|P_0|g_{\text{mod}}\rangle_{\text{true}}| \ll \delta, \tag{11}$$

where $\delta$ is the accuracy scale set in the task.

# 3 Protocol to extract information

We now describe how the observers near the boundary can complete the task described in section 2, using the manipulations and measurements described there. First we describe the main idea, and then describe a more detailed algorithm that covers some subtleties and exceptional cases.

## 3.1 The main idea

In describing the main idea, we will make certain simplifying assumptions. However, we emphasize that our procedure is completely general, and in the next subsection we account for all possible special cases.

For simplicity, consider a state where $\langle 0|g\rangle = 0$, i.e. the state that the observers are given has no overlap with the vacuum itself. If the state is not of this form to start with, we explain below how the observers can perform a simple preliminary manipulation to ensure this. Now consider the combined effect of acting with a unitary (as displayed in (7)) followed by the measurement of the energy and a determination of the relative frequency with which this energy-measurement yields 0 as displayed in (10).

---

[1]This can be seen by straightforwardly quantizing fields about global AdS and examining the low-energy spectrum. From the perspective of AdS/CFT, we note that the spectrum of energy levels in global AdS is dual to the spectrum of operator-dimensions in the boundary theory, which is expected to be discrete.

This relative frequency is easy to compute in perturbation theory. Using equation (8), we see that the relative frequency with which the measurement of the energy yields 0 is then given by

$$\langle g_{\text{mod}}|P_0|g_{\text{mod}}\rangle = J^2|\langle 0|X|g\rangle|^2 + \text{O}\left(J^3\right) = J^2|\langle X|g\rangle|^2 + \text{O}\left(J^3\right). \tag{12}$$

Note that the second order term above arises from combining two of the $\text{O}(J)$ terms displayed in Eqn. (8). Also, as advertised above, we note that we were justified in neglecting the $\text{O}(J^2)$ term in Eqn. (8); that term does not enter into the leading result above, since $\langle 0|g\rangle = 0$. We draw the reader's attention to the notation that we have used

$$|X\rangle = X|0\rangle, \tag{13}$$

since we will use the same notation multiple times below.

As described in Appendix A, if $X$ varies over the set of low-energy Hermitian operators that are localized near the boundary, then the set of states (13) already yields a complete basis for the low-energy Hilbert space. This may sound like an unfamiliar statement, but this is simply related to the *state-operator map* that is familiar from the study of quantum field theories. We emphasize that this is *not* just a formal property of the theory. In Appendix A we show how to construct the operators dual to a particular state explicitly.[2]

The upshot is that *the simple physical process described above is already sufficient to tell us the absolute value of the overlap of* $|g\rangle$ *with a set of basis vectors in the theory.*

Now this process by itself is not sufficient to tell us the *phase* of this overlap. But we can determine the phase as follows. Say that the observers have two *reference* Hermitian operators, $X_r$ and $X_i$. These operators have the property that the matrix elements of these operators between the state $|g\rangle$ and the vacuum are both non-zero and purely real and purely imaginary respectively.

$$\langle g|X_r\rangle = \text{Re}\left(\langle g|X_r\rangle\right) \neq 0; \qquad \langle g|X_i\rangle = i\text{Im}\left(\langle g|X_i\rangle\right) \neq 0. \tag{14}$$

Here, as above, we have used the notation $|X_r\rangle = X_r|0\rangle$ and likewise for $|X_i\rangle$. We explain below how the observers can generate such reference operators without much difficulty.

Then the phase of $\langle g|X\rangle$ can be fixed easily for all other operators. The observers first act with the unitary

$$U = 1 + iJ(X_r + X) + \text{O}\left(J^2\right), \tag{15}$$

and then by measuring the energy, as above, they determine the expectation value of $P_0$ in the modified state to second order in $J$.

This measurement allows the observers to determine $|\langle g|X_r\rangle + \langle g|X\rangle|^2$ through direct measurement. But note

$$\text{Re}\left(\langle g|X\rangle\right) = \frac{|\langle g|X_r\rangle + \langle g|X\rangle|^2 - \langle g|X_r\rangle^2 - |\langle g|X\rangle|^2}{2\langle g|X_r\rangle}. \tag{16}$$

Since all terms on the right hand side are known, the observers can use this to determine $\text{Re}\left(\langle g|X\rangle\right)$. This still leaves a *sign ambiguity* in $\text{Im}\left(\langle g|X\rangle\right)$. This can be fixed by acting with the unitary

$$U = 1 + iJ(X_i + X) + \text{O}\left(J^2\right), \tag{17}$$

---

[2]We have found that this point often causes confusion, and so we would like to repeat that the set of states in (13) would form a basis for the Hilbert space even in a QFT without gravity. But this property by itself would *not* enable the observers near the boundary to obtain any information about the behaviour of the state $|g\rangle$ in the bulk in a local QFT. In gravity we are assisted by the fact that the process of measuring the energy, and determining the frequency with which the measurement yields 0, corresponds, mathematically, to the insertion of a one-dimensional projector $P_0$ in (12). In non-gravitational theories, including gauge-theories, as we explain in section 4, the measurement of any operator near the boundary always corresponds to an infinite-dimensional projector, and so the near-boundary observers can obtain *no* information about the bulk.

and measuring $P_0$ in the modified state. The observers use this to determine $|\langle g|X_i\rangle + \langle g|X\rangle|^2$ and then note that

$$\text{Im}\left(\langle g|X\rangle\right) = \frac{|\langle g|X_i\rangle + \langle g|X\rangle|^2 - |\langle g|X_i\rangle|^2 - |\langle g|X\rangle|^2}{2\text{Im}\left(\langle g|X_i\rangle\right)}. \tag{18}$$

Note that the left hand sides of equations (16) and (18) are subject to a strong consistency check: upon squaring and adding, they must yield $|\langle g|X\rangle|^2$, which is known independently.

By using this procedure, the observers can determine the overlap of $|g\rangle$ with any state of the form (13). Since such states form a basis, this completely determines $|g\rangle$.

## 3.2 Details of the protocol

We now fill in some of the details that we omitted above, address various possible exceptions, and explain how to generate the reference operators used above. We start by explaining how the observers can perform an initial manipulation on the state to remove its overlap with the vacuum.

### 3.2.1 Preliminary step: Determination of $|\langle 0|g\rangle|^2$ and initial manipulation

First, the observers make a number of measurements of the energy on their identically prepared systems, without performing any manipulation at all. The probability that they obtain the answer 0 is given by

$$\langle g|P_0|g\rangle = |\langle 0|g\rangle|^2. \tag{19}$$

By performing a sufficient number of measurements so that the relative frequency tends to the probability, they are able to determine $|\langle 0|g\rangle|^2$ to any desired accuracy. If $|\langle 0|g\rangle|^2 = 0$, the observers proceed to the next step.

Otherwise, the observers only need to perform a simple manipulation of the state. They act with a simple *local unitary* that takes

$$|g\rangle \to \mathfrak{U}^z|g\rangle, \tag{20}$$

so that $\langle 0|\mathfrak{U}^z|g\rangle = 0$. The construction of this unitary is described in greater detail in Appendix B. But the main idea is very simple: it is easy to make two states in a large Hilbert space orthogonal by altering the state of a single degree of freedom. Here, let $O(t, \Omega)$ be the boundary value of a bulk propagating field as displayed in (4). By smearing this field with two suitably chosen functions, $f_1, f_2$ which have limited support *both* in time — so that they vanish outside $t \in [0, \epsilon]$ — and in space — so that they vanish outside a small region on the sphere — we can find operators that satisfy the Heisenberg algebra and thereby isolate a simple-harmonic degree of freedom. The local unitary described in Appendix B acts only on this simple harmonic degree of freedom. This is already sufficient to make the state $\mathfrak{U}^z|g\rangle$ orthogonal to the vacuum. The unitary may inject some energy into the state, but the state remains a low-energy state.

Note that if the observers can reconstruct the state $\mathfrak{U}^z|g\rangle$, since they know the unitary, $\mathfrak{U}^z$, they can back-calculate the state $|g\rangle$. In the discussion below, we will assume that this unitary operation has been performed. Rather than introducing separate notation for the case where $\langle 0|g\rangle = 0$ from the start, and for the case where the observes are required to act with an additional unitary, we will simply assume that $\langle 0|g\rangle = 0$ in all equations below.

### 3.2.2 Determination of reference operators

We now show how the observers can find two operators that satisfy the relations (14).

The operator $X_r$ is particularly easy to find. By trial and error and by using the protocol described above, the observers need to find only one Hermitian operator with the property that $|\langle g|X_r\rangle|^2 \neq 0$. Since the *overall phase* of $|g\rangle$ is physically meaningless, the observers can immediately choose the convention that

$$\langle g|X_r\rangle = |\langle g|X_r\rangle|, \tag{21}$$

which satisfies the first part of (14).

The observers can now find the operator $X_i$ as follows. Consider the set of all states at a given energy. We denote these states by $|E, \{\ell\}\rangle$ where we have separated the energy eigenvalue, $E$, from other possible quantum-numbers of the state that we have clubbed into $\{\ell\}$. Then, for each state at this energy, it is always possible to find two Hermitian operators, $X_{E,\{\ell\}}$ and $Y_{E,\{\ell\}}$ near the boundary, which have the property that

$$\left(X_{E,\{\ell\}} + iY_{E,\{\ell\}}\right)|0\rangle = |E, \{\ell\}\rangle. \tag{22}$$

Note that the operator on the left-hand side is *not* Hermitian due to the factor of $i$ that multiplies $Y_{E,\{\ell\}}$.

We pause to mention an important physical point. The observers do not need to find the exact operators $X_{E,\{\ell\}}$ and $Y_{E,\{\ell\}}$ that satisfy equation (22). It is acceptable for them to attain a level of accuracy that is controlled by the error $\delta$ that appears as part of the task-specification in equation (6). In particular, the explicit construction of such operators in Appendix A does not keep track of $O\left(\frac{1}{N}\right)$ corrections, which is not a problem because $\delta \gg \frac{1}{N}$.

Now, by manipulating the state with various unitaries, as indicated in the table below, the observers can obtain a number of physical quantities.

| Unitary Manipulation | Quantity Inferred |
|---|---|
| $1 + iJX_{E,\{\ell\}} + O\left(J^2\right)$ | $\left|\langle g|X_{E,\{\ell\}}\rangle\right|^2$ |
| $1 + iJY_{E,\{\ell\}} + O\left(J^2\right)$ | $\left|\langle g|Y_{E,\{\ell\}}\rangle\right|^2$ |
| $1 + iJ\left(X_{E,\{\ell\}} + X_r\right) + O\left(J^2\right)$ | $\mathrm{Re}\left(\langle g|X_{E,\{\ell\}}\rangle\right)$ and $\left|\mathrm{Im}\left(\langle g|X_{E,\{\ell\}}\rangle\right)\right|$ |
| $1 + iJ\left(Y_{E,\{\ell\}} + X_r\right) + O\left(J^2\right)$ | $\mathrm{Re}\left(\langle g|Y_{E,\{\ell\}}\rangle\right)$ and $\left|\mathrm{Im}\left(\langle g|Y_{E,\{\ell\}}\rangle\right)\right|$ |
| $1 + iJ\left(X_{E,\{\ell\}} + Y_{E,\{\ell\}}\right) + O\left(J^2\right)$ | $\mathrm{Im}\left(\langle g|X_{E,\{\ell\}}\rangle\right)\mathrm{Im}\left(\langle g|Y_{E,\{\ell\}}\rangle\right)$ |

$$\tag{23}$$

In some cases, the quantity on the right column in the table above may be related to the direct observable through some simple algebra. For instance, in the last line above, we have

$$\begin{aligned}|\langle g|X_{E,\{\ell\}}\rangle + \langle g|Y_{E,\{\ell\}}\rangle|^2 = |\langle g|X_{E,\{\ell\}}\rangle|^2 + |\langle g|Y_{E,\{\ell\}}\rangle|^2 + 2\mathrm{Re}(\langle g|X_{E,\{\ell\}}\rangle)\mathrm{Re}(\langle g|Y_{E,\{\ell\}}\rangle) \\ + 2\mathrm{Im}(\langle g|X_{E,\{\ell\}}\rangle)\mathrm{Im}(\langle g|Y_{E,\{\ell\}}\rangle).\end{aligned} \tag{24}$$

The observers can determine the product of the imaginary parts since they know all other terms in the equation above: they obtain the left hand side of the equation through direct measurement and know the other terms on the right from previous measurements.

The table above leaves the observers with an ambiguity in the *sign* of the imaginary part of the overlap of $|g\rangle$ with the various basis vectors. This is because, in each case they know the real part and only the magnitude of the overlap. But since the last line in the table tells them about the product of the imaginary parts, this is a *correlated* ambiguity. Once they infer the sign of the imaginary part of a single overlap, they can immediately infer the signs of all the other imaginary parts.

This single sign can be fixed as follows. Through a measurement of the energy, the observers can also determine

$$\langle g|P_E|g\rangle = \sum_{\{\ell\}}|\langle g|E, \{\ell\}\rangle|^2, \tag{25}$$

where the sum runs over all states at that energy. The physical process for determining this is simply to measure the energy in the state $|g\rangle$ and determine the relative frequency with which the result $E$ is obtained. This directly yields the left hand side. But we have

$$\sum_{\{\ell\}}|\langle g|E,\{\ell\}\rangle|^2 = \sum_{\{\ell\}}|\langle g|X_{E,\{\ell\}}\rangle|^2 + |\langle g|Y_{E,\{\ell\}}\rangle|^2 + 2C_E, \tag{26}$$

where we have defined

$$C_E = \sum_{\ell}\Big(\mathrm{Re}\langle g|Y_{E,\{\ell\}}\rangle\mathrm{Im}\langle g|X_{E,\{\ell\}}\rangle - \mathrm{Re}\langle g|X_{E,\{\ell\}}\rangle\mathrm{Im}\langle g|Y_{E,\{\ell\}}\rangle\Big). \tag{27}$$

As we pointed out above, the signs of the all the imaginary parts are correlated. Therefore simply from the measurements in (23), we already know $|C_E|$. From the measurement of $\langle g|P_E|g\rangle$ we can fix the sign of $C_E$ and this immediately fixes the sign of all the imaginary parts.

The reference operator $X_i$ can then be generated using any operator with a non-zero imaginary part in its matrix element between $|g\rangle$ and the vacuum. If $X_{E,\{\ell\}}$ is such an operator,

$$X_i = X_{E,\{\ell\}} - \frac{\mathrm{Re}\big(\langle g|X_{E,\{\ell\}}\rangle\big)}{\langle g|X_r\rangle}X_r. \tag{28}$$

### 3.2.3 Complete protocol

Once the observers have determined a set of reference operators, they can then proceed to systematically determine the overlap of $|g\rangle$ with any set of basis states as explained in section 3.1.

For instance, they may choose to use the basis of energy eigenstates. For each energy eigenstate, they find the operator dual to it so that it can be written in the form (22). They then perform the deformations of the Hamiltonian given in the Table in (23). In addition, they require a manipulation by the following unitary

$$U = 1 + iJ(X_i + X_{E,\{\ell\}}) + \mathrm{O}\big(J^2\big). \tag{29}$$

A measurement of the energy following this unitary then immediately allows the observers to read off $\mathrm{Im}\big(\langle g|X_{E,\ell}\rangle\big)$ as explained near equation (18).

Together with the other physical quantities obtained by the deformations displayed in Equation (23), this allows the observers to *completely determine* $\langle g|E,\{\ell\}\rangle$ — both in magnitude and in phase. Proceeding in this manner for each separate energy eigenstate, below the UV-cutoff $\Lambda$, the observers completely determine the state $|g\rangle$ to the desired accuracy. A flowchart describing the entire process, including the verification described in the next section is shown in Figure 2.

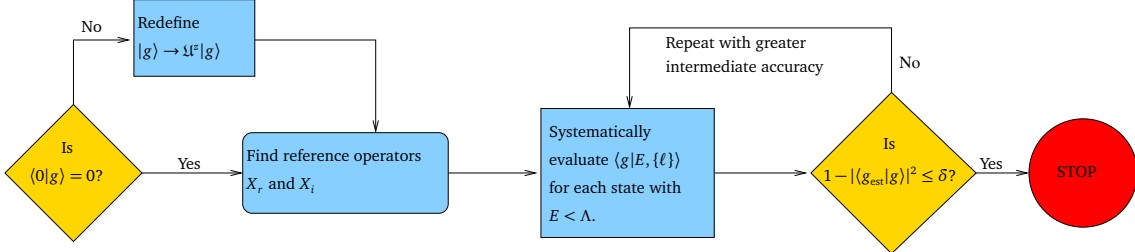

Figure 2: *A flowchart describing the key steps in our protocol.*

We would like to emphasize two obvious points.

1. There are various degeneracies in the energy-spectrum, but the observers can independently determine the overlap with each separate energy eigenstate. Each eigenstate is associated with a unique set of Hermitian operators $X_{E,\{\ell\}}$ and $Y_{E,\{\ell\}}$.

2. The observers can successfully perform their procedure, even if two energy eigenstates are related by a *global symmetry*. Since the states are distinct, the pair of Hermitian operators associated with the two energy eigenstates by equation (22) are also distinct. We give an explicit example in section 3.3.

### 3.2.4 Verification

In principle, it is sufficient for the observers to determine the overlap of the state with all energy eigenstates. However, since this involves a number of operations, errors may accumulate in this process. So we now explain how the observers can also verify that they have successfully completed the task, in only a few steps.

At the end of the process above, the observers have the following estimate of the state

$$|g_{\text{est}}\rangle = (X_{\text{est}} + iY_{\text{est}})|0\rangle, \tag{30}$$

where

$$
\begin{aligned}
X_{\text{est}} &= \sum_{E,\{\ell\}} \text{Re}\left(\langle E, \{\ell\}|g\rangle\right) X_{E,\{\ell\}} - \text{Im}\left(\langle E, \{\ell\}|g\rangle\right) Y_{E,\{\ell\}} \\
Y_{\text{est}} &= \sum_{E,\{\ell\}} \text{Im}\left(\langle E, \{\ell\}|g\rangle\right) X_{E,\{\ell\}} + \text{Re}\left(\langle E, \{\ell\}|g\rangle\right) Y_{E,\{\ell\}}.
\end{aligned}
\tag{31}
$$

Now the observers act with a two-parameter deformation of the Hamiltonian

$$U = 1 + i\left(J_1 X_{\text{est}} + J_2 Y_{\text{est}}\right) + \text{O}\left(J_1^2\right) + \text{O}\left(J_2^2\right), \tag{32}$$

followed by an energy measurement and a determination of the frequency with which this yields 0. As explained above, by determining the $\text{O}\left(J_1^2\right), \text{O}\left(J_2^2\right), \text{O}(J_1 J_2)$ terms in this observation, the observers obtain the values of

$$\alpha = |\langle g|X_{\text{est}}\rangle|^2; \quad \beta = |\langle g|Y_{\text{est}}\rangle|^2; \quad \gamma = \text{Re}\left(\langle g|X_{\text{est}}\rangle\right)\text{Re}\left(\langle g|Y_{\text{est}}\rangle\right) + \text{Im}\left(\langle g|X_{\text{est}}\rangle\right)\text{Im}\left(\langle g|Y_{\text{est}}\rangle\right). \tag{33}$$

Now with $\alpha, \beta, \gamma$ defined as above

$$|\langle g|X_{\text{est}}\rangle + i\langle g|Y_{\text{est}}\rangle|^2 = \alpha + \beta \pm 2\sqrt{\alpha\beta - \gamma^2}. \tag{34}$$

The observers can easily fix the sign of the square-root. The first check is that with one of the two possible signs, the observers should obtain exactly 1 for the right hand side. This is already an extremely strong check on the estimates of the observers. But the observers can additionally verify that this is the correct sign by using the reference operators above to independently determine the real and imaginary parts of the overlap between $|g\rangle$ and $|X_{\text{est}}\rangle$ and $|Y_{\text{est}}\rangle$.

## 3.3 An example with global symmetries

In the discussion above, we have *not* assumed the absence of global symmetries. It may be, for other reasons [29,30], that global symmetries do not exist in a theory of quantum gravity. But this issue does not affect our protocol. The main physical point is that our protocol involves not only measurements of the energy but also of *correlators of the Hamiltonian with other*

*dynamical fields*. These correlators can break the degeneracy between states related by global symmetries.

To demonstrate this, we now give an example of how the observers can identify the state, in a situation where the bulk theory does have global symmetries. In addition to the field of mass $m$ dual to a boundary operator as displayed in equation (4), say that we have another field $\widetilde{\phi}$ of exactly the same mass, dual to a boundary operator $\widetilde{O}$.

Then the low-energy theory contains two states of the same energy: $\frac{d}{2} + \sqrt{(\frac{d}{2})^2 + m^2}$. Denoting the normalized states by $|\Delta\rangle$ and $|\widetilde{\Delta}\rangle$, we put the system in a state

$$|g\rangle = a|\Delta\rangle + b|\widetilde{\Delta}\rangle, \tag{35}$$

with $|a|^2 + |b|^2 = 1$. The task of the observers is to determine the complex number $\frac{b}{a}$. (The overall phase of the state is meaningless.)

Both energy eigenstates can be written as

$$|\Delta\rangle = (X_\Delta + iY_\Delta)|0\rangle, \qquad |\widetilde{\Delta}\rangle = (\widetilde{X}_\Delta + i\widetilde{Y}_\Delta)|0\rangle. \tag{36}$$

It is possible to find *explicit* real functions $f_R(t,\Omega)$ and $f_I(t,\Omega)$, as explained in Appendix A, that are supported on $t \in [0,\epsilon]$ and satisfy

$$X_\Delta = \int O(t,\Omega)f_R(t,\Omega)dt d^{d-1}\Omega; \qquad Y_\Delta = \int O(t,\Omega)f_I(t,\Omega)dt d^{d-1}\Omega$$
$$\widetilde{X}_\Delta = \int \widetilde{O}(t,\Omega)f_R(t,\Omega)dt d^{d-1}\Omega; \qquad \widetilde{Y}_\Delta = \int \widetilde{O}(t,\Omega)f_I(t,\Omega)dt d^{d-1}\Omega. \tag{37}$$

Due to the global symmetry, the same functions $f_R$ and $f_I$ appear for both states in the right hand side of the equation above, but notice that $|\Delta\rangle$ is produced by applying $O$ to the vacuum, whereas $|\widetilde{\Delta}\rangle$ is produced by applying $\widetilde{O}$ to the vacuum.

As a result of the global symmetry, the states generated by the operators above satisfy

$$\langle X_\Delta | Y_\Delta \rangle = \langle \widetilde{X}_\Delta | \widetilde{Y}_\Delta \rangle; \qquad \langle X_\Delta | X_\Delta \rangle = \langle \widetilde{X}_\Delta | \widetilde{X}_\Delta \rangle; \qquad \langle Y_\Delta | Y_\Delta \rangle = \langle \widetilde{Y}_\Delta | \widetilde{Y}_\Delta \rangle;$$
$$\langle X_\Delta | \widetilde{X}_\Delta \rangle = \langle X_\Delta | \widetilde{Y}_\Delta \rangle = \langle Y_\Delta | \widetilde{X}_\Delta \rangle = \langle Y_\Delta | \widetilde{Y}_\Delta \rangle = 0. \tag{38}$$

Using the protocol above, the observers can first find the ratio of the magnitudes of the coefficients:

$$\frac{|b|}{|a|} = \frac{|\langle \widetilde{X}_\Delta | g \rangle|}{|\langle X_\Delta | g \rangle|}. \tag{39}$$

Second, they choose the convention for the overall phase of $|g\rangle$ so that $\langle X_\Delta | g \rangle = |\langle X_\Delta | g \rangle|$. This is equivalent to fixing

$$a = \frac{|\langle X_\Delta | g \rangle|}{\langle X_\Delta | \Delta \rangle}. \tag{40}$$

The phase of $b$ can also be found using the operator $X_\Delta$, from (37), in the role of the reference operator $X_r$. (The reference operator $X_i$ will not be required in this case.) Using the method above, the observers can determine the values of

$$\text{Re}(\langle \widetilde{X}_\Delta | g \rangle) = \text{Re}(b\langle \widetilde{X}_\Delta | \widetilde{\Delta} \rangle); \quad \text{Re}(\langle \widetilde{Y}_\Delta | g \rangle) = \text{Re}(b\langle \widetilde{Y}_\Delta | \widetilde{\Delta} \rangle). \tag{41}$$

Since we already know the magnitude of $b$, the two equations above unambiguously tell us the phase of $b$, and therefore the complex number $\frac{b}{a}$ as required.

# 4 Local quantum field theories

In section 3 we explained how, in a theory of gravity, the near-boundary observers could identify the bulk state. In this section, we explain that in a local quantum field theory in AdS, not only are the observers *unable* to identify the bulk state, their ignorance about the state near the middle of AdS is *complete*. This is a consequence of the so-called "split property" of local quantum field theories that we review below. We start with a discussion of QFTs without any gauge fields and then include gauge fields.

## 4.1 Local QFTs without gauge fields

In a local QFT, our treatment can be a little more rigorous because we no longer need to make any distinction between simple and complicated operators. Let $\phi_i(t, r, \Omega)$ be the set of local quantum fields with the boundary conditions that

$$\phi_i(t, r, \Omega) \xrightarrow[r \to \infty]{} \frac{1}{r^{\Delta_i}} O_i(t, \Omega). \tag{42}$$

Then we define two algebras. The first is

$$\mathcal{A}_{[0,\epsilon]} = \text{span of} \{ O_{i_1}(t_1, \Omega_1) \ldots O_{i_n}(t_n, \Omega_n) \}, \qquad t_i \in [0, \epsilon], \tag{43}$$

with no constraint on the coordinates $\Omega_i$ and no limit on $n$. In a local QFT, this algebra is precisely the same as the algebra of bulk fields on the time-slice $t = \frac{\epsilon}{2}$ with the radial coordinate in the range $r \in [\cot\left(\frac{\epsilon}{2}\right), \infty)$. We can also define an algebra of commuting operators

$$\mathcal{A}_{\text{bulk}} = \text{span of} \{ \phi_{i_1}(t = \frac{\epsilon}{2}, r_1, \Omega_1) \ldots \phi_{i_n}(t = \frac{\epsilon}{2}, r_n, \Omega_n) \}, \qquad r_i < \cot(\frac{\epsilon}{2}) - \chi. \tag{44}$$

Here $\chi$ is a small parameter that separates the causal wedge of the time band $[0, \epsilon]$ from the support of the operators that are elements of $\mathcal{A}_{\text{bulk}}$. The support of the two algebras is shown in Figure 3.

Note that the support of the operators in $\mathcal{A}_{\text{bulk}}$ is in a region that is spacelike to the time-band $[0, \epsilon]$ on the boundary. Therefore, by microcausality, we have

$$[A_1, A_2] = 0, \qquad \forall A_1 \in \mathcal{A}_{[0,\epsilon]}, A_2 \in \mathcal{A}_{\text{bulk}}. \tag{45}$$

We give the near-boundary observers the same task as in section 2. Their powers are that they are allowed to act with an arbitrary unitary from $\mathcal{A}_{[0,\epsilon]}$ and make arbitrary projective measurements from the same algebra.

First, we note an obvious point. Let $P_1, U_1 \in \mathcal{A}_{[0,\epsilon]}$ be, respectively, an arbitrary projector and arbitrary unitary from the algebra of the time band. Let $U_{\text{bulk}} \in \mathcal{A}_{\text{bulk}}$ be an arbitrary unitary from the commuting algebra. Then we have

$$\langle g | U_1^\dagger P_1 U_1 | g \rangle = \langle g | U_{\text{bulk}}^\dagger U_1^\dagger P_1 U_1 U_{\text{bulk}} | g \rangle. \tag{46}$$

This is an exact relation by microcausality. So the observers cannot distinguish the state $|g\rangle$ from $U_{\text{bulk}} |g\rangle$ by any combination of possible manipulations and measurements that they are allowed to make.

The reader might wonder if the observers can get at least "some" information about the bulk state. But even this turns out to be impossible by virtue of the so-called *split property* [31]. The split property can be phrased as follows. Let $|\Psi_1\rangle$ and $|\Psi_2\rangle$ be two arbitrary states in the Hilbert space. Then the split property tells us that it is possible to find a state $|g\rangle$, so that

$$\langle g | A_1 A_2 | g \rangle = \langle \Psi_1 | A_1 | \Psi_1 \rangle \langle \Psi_2 | A_2 | \Psi_2 \rangle, \qquad \forall A_1 \in \mathcal{A}_{[0,\epsilon]}; A_2 \in \mathcal{A}_{\text{bulk}}. \tag{47}$$

This tells us that it is possible to find a global state $|g\rangle$ with the following properties.

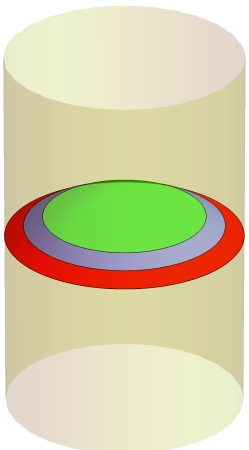

Figure 3: *In a local QFT, the algebra $\mathcal{A}_{bulk}$ supported in the inner region (green) exactly commutes with the algebra $\mathcal{A}_{[0,\epsilon]}$ supported in the outermost region (red). The "split property" of local QFTs tells us that when the regions are separated by a small "collar" (blue region), the wavefunctions of the two regions can be prepared absolutely independently. The discussion of section 3 tells us that in a theory with dynamical gravity, split states do not exist for the configuration above.*

1. For all observations made near the boundary, $|g\rangle$ looks exactly like $|\Psi_1\rangle$.

2. For all observations made in the bulk, $|g\rangle$ looks exactly like $|\Psi_2\rangle$.

3. The results of observations made jointly near the boundary and in the bulk are *uncorrelated*.

Split states can be constructed explicitly in local QFTS as described in [32].

We briefly contrast these results with gravity. The analysis of section 3 implies that, for the geometrical configuration of Figure 1b — where a region is completely surrounded by its complement — split states do not exist in gravity, at least within the low-energy Hilbert space. This is closely tied to the fact that there are no local gauge-invariant operators in gravity. So, unlike local QFTs, it is simply not possible to find a local unitary $U_{\text{bulk}}$ that commutes with operators near the boundary and changes the state in the interior of the spacetime.[3]

## 4.2 Non-gravitational gauge theories

We now turn to non-gravitational gauge theories. Very superficially, it might appear that in such theories the near-boundary observers could obtain some information about the bulk by taking advantage of the conserved charges that are defined by boundary integrals. However, this turns out not to be the case. In particular, the procedure of section 3 cannot be repeated at all because there is no analogue of the "projector on the vacuum" which projects onto a unique state. In contrast, the "projector onto states of zero charge" in a non-gravitational theory projects onto an *infinite-dimensional subspace* of states.

This is physically related to the fact that non-gravitational gauge theories have exactly local gauge-invariant operators. For instance, consider a small Wilson loop that is localized at time

---

[3]We believe that even in gravity, it should be possible to find split states for geometrical configurations where the region and its complement both include a part of the asymptotic boundary. This is consistent with the fact that, in such configurations, it is possible to find commuting algebras by dressing operators from the region and its complement to different parts of the asymptotic boundary. But this topic is beyond the scope of this paper.

$t = \frac{\epsilon}{2}$ and is confined in the radial coordinate to $r < \cot(\frac{\epsilon}{2}) - \chi$ as we discussed above. This Wilson loop furnishes an example of a unitary operator $U_{\text{bulk}}$ that commutes with *all* physical manipulations and measurements that can be made in the near-boundary region.

There is no unique way to identify the algebra associated with a region in gauge theories. This point was discussed extensively in [33–36]. Physically if one regulates the theory on a lattice, then the "link" variables cut through the boundary of any region. So one has to decide whether to count them as part of the region or not. The different possible choices lead to different centers for the algebra of the region.

However, since we have a "collar region" that separates the algebra inside from the algebra outside, the different choices of center should not affect the validity of equation (47), even in gauge theories. This means that once again, in the absence of gravity, the observers have no information about the state near the center of AdS.

Note that the observers outside can determine the charge in the union of the collar region and the interior region. This is the form in which the split property for gauge theories is stated in [37]. But since the collar can hold an arbitrary amount of charge, if one focuses attention only on the interior region, then the near-boundary observers have no information just as in a local QFT without gauge fields.

### 4.2.1 Special case: information with priors

We now discuss a special case where the observers are given strong priors about the possible global state. In this special case, the observers can use long-range gauge fields to obtain information about the interior. We caution the reader that both the task given to the observers and the prior information available to them in this problem is quite different from the discussion in the rest of the paper. So we urge the reader not to confuse the discussion in this subsection with the general discussion in the rest of the paper.

We again consider two bulk fields, $\phi$ and $\widetilde{\phi}$ with the same mass $m$ and boundary values $O$ and $\widetilde{O}$ as in section 3.3. The difference is that we switch off dynamical gravity but we gauge the global $SO(2)$ symmetry. We also fix a gauge so that it is meaningful to speak of the value of the fields at a bulk point $\phi(t, r, \Omega)$. When the symmetry is gauged, the charge, $Q$, is an element of the algebra $A_{[0,\epsilon]}$ with commutation relations

$$[Q, \phi(t, r, \Omega)] = i\widetilde{\phi}(t, r, \Omega); \quad [Q, \widetilde{\phi}(t, r, \Omega)] = -i\phi(t, r, \Omega). \tag{48}$$

We now consider the situation where the observers are told *ahead of time* that the global state is of the form

$$|g\rangle = \exp\left[i\lambda \int \left(f(r, \Omega)\phi(t = \frac{\epsilon}{2}, r, \Omega) + \widetilde{f}(r, \Omega)\widetilde{\phi}(t = \frac{\epsilon}{2}, r, \Omega)\right) dr d^{d-1}\Omega\right]|0\rangle, \tag{49}$$

where $f(r, \Omega)$ and $\widetilde{f}(r, \Omega)$ have support for $r < \cot\frac{\epsilon}{2} - \chi$. Moreover, we allow the observers to explore this state for different values of $\lambda$ near $\lambda = 0$. Note that, by fiat, we have disallowed the action of additional bulk unitaries on the state. The observers are given the task of determining the real functions $f(r, \Omega)$ and $\widetilde{f}(r, \Omega)$.

The observers now act with a unitary

$$U = 1 + iJ \int \left[O(t, \Omega)h(t, \Omega) + \widetilde{O}(t, \Omega)\widetilde{h}(t, \Omega)\right] dt d^{d-1}\Omega + O(J^2). \tag{50}$$

In the modified state, the observers measure the global charge and compute the expectation value of $Q^2$ to first order in $J$ and to first order in $\lambda$.

A simple calculation yields

$$\langle g_{\text{mod}}|Q^2|g_{\text{mod}}\rangle = J\lambda\left(\langle f,h\rangle + \langle \widetilde{f},\widetilde{h}\rangle\right) + \dots, \tag{51}$$

where ... denotes higher order terms and the inner-product $\langle,\rangle$ is defined through

$$\langle f,h\rangle \equiv \int \langle 0|\{\phi(t=\frac{\epsilon}{2},r,\Omega),O(t',\Omega')\}|0\rangle f(r,\Omega)h(t',\Omega')d^{d-1}\Omega d^{d-1}\Omega' dr dt'$$
$$= \mathcal{N}_\Delta \int f(r,\Omega)h(t',\Omega')\left[\frac{1}{\sqrt{1+r^2}\cos(t'-\frac{\epsilon}{2})-r\Omega\cdot\Omega'}\right]^\Delta d^{d-1}\Omega d^{d-1}\Omega' dr dt'. \tag{52}$$

In the second line above we have substituted the explicit form of the two-point function, and $\mathcal{N}_\Delta$ is an unimportant numerical factor.

The reader will notice that something interesting has happened. On the right hand side above, we have a convolution, using the bulk to boundary propagator, of the function $f$ with $h$ and separately of the function $\widetilde{f}$ with $\widetilde{h}$. Since the observers can choose any pair of functions $h, \widetilde{h}$ with support in the time-band $t' \in [0,\epsilon]$, this is sufficient to completely reconstruct $f$.

We will show this by proving that that there is no non-zero function $f$ so that $\langle f,h\rangle = 0$, for all functions $h$ with support in $t \in [0,\epsilon]$. This inner-product is just the real part of the overlap of the bra $\int \langle 0|\phi(t=\frac{\epsilon}{2},r,\Omega)f(r)dr d\Omega$ with the ket $\int O(t',\Omega')h(t',\Omega')dt'd\Omega'|0\rangle$. In fact the overlap has no imaginary part because the operators are spacelike separated.[4] But, by the arguments of Appendix A, it is impossible to choose $f$ to make this overlap vanish for all possible choices of $h$.

It is instructive to understand what is happening in this example. The operators $\phi$ and $\widetilde{\phi}$ are not local operators since, in a gauge-invariant formalism, they would have Wilson lines stretching out to the boundary. The procedure above took advantage of these Wilson-line "tails" to extract the functions $f$ and $\widetilde{f}$. The procedure was successful because the observers were given the *prior* that the state was of the form (49). Without the prior, the observers could have made no progress because, as explained above, in a non-gravitational gauge theory, it *is* possible to hide information from the near-boundary observers by acting with local gauge invariant operators in the bulk.

However, in the gravitational setting, since there are no local gauge-invariant operators, one cannot change the state in the interior without having some effect that propagates out to the boundary. This is the essential reason that in gravitational theories, the observers can determine the state in the bulk even without a prior of the form (49).

# 5 Conclusion and discussion

**Main result**

Our main result is that in a theory of quantum gravity in global AdS, observers near the boundary can extract information about the bulk through a physical process. This result is closely tied to the arguments in [1,2]. These papers argued that in theories of gravity — with either asymptotically AdS or asymptotically flat boundary conditions — operators that probe the bulk have a dual representation as operators near the boundary. The innovation in this paper is that we have provided a *physical* protocol by means of which bulk information can be extracted.

---

[4]While bulk operators like $\phi$ may fail to commute with boundary operators like $O$ even at spacelike separation due to the Wilson lines that stretch from $\phi$ to the boundary, this effect appears only at the next order in perturbation theory in the two-point function.

Our analysis assumed that observers near the boundary could perform the standard operations that are allowed in quantum mechanics — manipulations of the system through unitary operators and projective measurements. The unitary operators that the observers use in our protocol are in one to one correspondence with low-energy Hermitian operators in the near-boundary region. The key part of our protocol is that, in a theory of gravity, the observers can follow such a unitary with a measurement of the energy, and a determination of the relative frequency with which this measurement yields 0. By the logic sketched in the introduction, and then explained in greater detail in section 3, the observers can use this procedure to determine the state in the bulk.

**Implications for quantum information measures**

The von Neumann entropy, which is a commonly used quantum-information measure, precisely measures the uncertainty in the state that remains after an observer has extracted all possible information that can be obtained through manipulations by local unitaries and local measurements. But what we have shown here is that when the global state of the system is a low-energy state, the near-boundary observers can identify the state precisely with no uncertainty.

In a theory of gravity, since the spacetime can fluctuate, one must be careful about what one means by the von Neumann entropy of a region. One possibility is to define a region geometrically using a relational prescription and then consider quantum-information measures defined with respect to a set of simple operations confined to that region [19]. With this definition in mind, consider the region defined by the set of points $\{r, t = \frac{\epsilon}{2}, \Omega\}$ with $r > r_0$ and $\Omega$ arbitrary. This is the shaded annular region shown in Figure 1b with $r_0 = \cot(\frac{\epsilon}{2})$. This can be defined relationally as the causal wedge of the boundary time-band with $t \in [0, \epsilon]$. Then our analysis suggests that the von Neumann entropy of this region is 0. This is very different from the same quantity, as computed in a local quantum field theory, where one might expect an answer proportional to $r_0^{d-1}$ after UV-regulation. This provides a striking example of how the von Neumann entropy is *not* a perturbative quantity. Turning on weak gravitational effects changes this quantity from a finite value to 0. It would be interesting to understand the relationship of this result with the results of [38].

A similar comment holds for the relative entropy, which quantifies how well two states can be distinguished using observations in a region. We can consider the relative entropy of the states on an annular region of the form above as a function of $r_0$. In a local quantum field theory, by the monotonicity of relative entropy, we would expect this answer to increase as $r_0$ decreases. The result of our analysis is that since the information content does not increase in a theory of gravity, the relative entropy is *constant* as a function of $r_0$.

It may be possible to find mathematical generalizations of the von Neumann entropy or relative entropy in the presence of gravity by generalizing the replica trick [39], or through some other method. But, if these generalizations are to have the usual physical interpretation in terms of the information available in a region then we believe that they should agree with the answers indicated above.

Also note that our result is *not* in contradiction with the RT [40, 41] or HRT formulas [42] or even their quantum corrected versions [22, 23]. In these setups, one always considers a *subregion* on the boundary dual to an entanglement wedge in the bulk. Our analysis is applicable to regions which include an entire time-slice of the boundary, and so it does not apply to cases where the region and its complement both include a part of the boundary.

**Failure of the split property in gravity**

A physical way to understand our result is that each bulk excitation leaves a distinctive imprint on the wavefunction near the boundary. This imprint can be read off by near-boundary

observers using a set of physical manipulations and measurements. This is quite different from a local quantum field theory since it means that it is *not* possible to modify the wavefunction inside a bounded region without also modifying the wavefunction outside it.

But this immediately means that the "split property" fails in gravity, at least for the geometrical configuration where a region is surrounded by its complement. It is sometimes incorrectly stated [43] that gravitational effects allow one to read off the expectation value of the energy and other Poincare charges, but do not yield information beyond this. But as the analysis in this paper shows, the constraints in gravity are far more powerful and allow one to obtain complete information about the interior.

We expect that the split property will be recovered if we consider a different kind of geometrical configuration, where the region and its complement both include some part of the asymptotic boundary.

**No superluminal communication**

A question that is often asked is as follows: what if an observer enters the bulk and sets off an "explosion" near the middle of AdS. Will the near-boundary observers not know about this on the same time-slice, and does this not lead to a protocol for superluminal communication? In fact, our protocol does not lead to a protocol for superluminal communication and this can be seen in two ways.

One obvious reason is that the near-boundary observers are spread out over the entire sphere at large $r$. In order to determine that an explosion has taken place, they need to travel around the sphere to combine their results and this itself takes at least as much time as the light-crossing time of AdS. So they cannot extract information about the explosion "any faster" than another set of observers who are allowed to enter the bulk and directly make bulk measurements.

But the second deeper reason is as follows. In a local quantum field theory, the precise way to check if an observer in the interior can send a signal faster than the speed of light to observers in the exterior is as follows. We consider two different wavefunctions of the theory so that at a given time, they differ inside some bounded region but are the same outside. We then allow the wavefunctions to evolve with time, and check if the region, within which they differ, grows faster than the speed of light. However, the failure of the split property alluded to above tells us that one cannot set up this experiment in gravity at all: if two wavefunctions differ inside, they will also differ outside. Said another way, the observer inside cannot transmit information superluminally to the near-boundary observers because the near-boundary region already contains a copy of all information from the interior!

We emphasize that it is still possible to ask questions about *asymptotic causality*. If we make a disturbance near the boundary at some point of time, it is still important to check that this disturbance does not travel to another part of the boundary through the bulk any faster than it can travel through the boundary. This kind of check was performed in [44] (see [45] for recent developments) and yields important constraints on bulk dynamics.

**Implications for black holes**

The analysis in this paper has explicitly focused on low-energy states. In the case of black holes, one can ask two kinds of questions. First, consider the set of states in the vicinity of a given microstate. This is sometimes called the "little Hilbert space" [20] and it includes excitations in the black hole interior [46]. Then one can ask if the observers can extend the protocol described here to uniquely identify the excitation. In effective field theory, the spectrum of possible excitations about a black hole appears to be continuous because the gaps between energy levels in the vicinity of a black-hole microstate are too small to be seen perturbatively. Nevertheless, the analysis of [47,48] suggests that our protocol can be extended to black holes,

as we will explore in forthcoming work.

However, a second kind of question is as follows: can the observers determine the microstate of the black hole itself from observations in the exterior? It is clear that this question cannot be answered within the framework of this paper where the observers are restricted to acting with simple unitaries. But this is not surprising. Any protocol to decode the microstate of the black hole — whether it involves making direct observations on the Hawking radiation or indirect observations through gravitational effects as described here — will necessarily require the observers to perform complex manipulations and very accurate measurements.

But, with the caveat above, it is interesting that — although such an analysis requires some careful extrapolations as discussed in [1] — it is, in principle, possible to extend this protocol all the way to black hole microstates.

**Possible obstructions to the protocol**

Our protocol leads to results that are in conflict with common intuitive notions of locality. So it is important to ask if a more careful consideration of the physics could reveal possible physical obstructions to the implementation of this protocol. We now sketch some possibilities in this direction.

The standard framework used in discussions of quantum information relies on a separation between the system and the observer. This is because both unitary manipulations and measurements require the observers to deform the Hamiltonian. We described in section 2 how our setup could be realized by embedding multiple copies of global AdS within a larger spacetime. But it may happen that this arrangement leads to subtleties. For instance, since the system is gravitational, the observers cannot decouple exactly from the system that they are observing.

We have also assumed that there is no obstruction to making measurements of the energy at the AdS scale. But it is possible that such measurements are difficult for some reason. Note that the specific issues discussed in [49] are *not* directly relevant for our protocol. The paper [49] was written in the context of flat space but even there, as described in Appendix B of [1], the Hamiltonian can be measured accurately by making smeared measurements over a region with large radial extent. This issue has not been studied in AdS, but even if such a smearing is required, it can be performed within the annular region of Figure 1b, which corresponds to an infinite range of the radial coordinate. But we cannot rule out the possibility that other effects in the "same universality class" are significant in our context.

We do not have any evidence that effects of the kind above are important. But it would be extremely interesting if they are, since this would teach us that, in gravitational systems and in cosmological settings, one must be cautious while using the standard rules for observables in quantum mechanics.

# Acknowledgments

S.R. is partially supported by a Swarnajayanti fellowship DST/SJF/PSA-02/2016-17 of the Department of Science and Technology. We are very grateful to Kyriakos Papadodimas for collaboration in the early stages of this work. We are grateful to Bidisha Chakraborty, Sumit Das, Victor Godet, Anna Karlsson, Alok Laddha, R. Loganayagam, Gautam Mandal, Juan Maldacena, Shiraz Minwalla, Mehrdad Mirbabayi, Siddharth Prabhu, Pushkal Shrivastava, and Sandip Trivedi for several useful discussions.

# A  State-operator map

In this appendix, we would like to describe, in some more detail, how states of the form

$$|X\rangle = X|0\rangle,\tag{53}$$

form a basis for the entire Hilbert space. Here $X$ is a simple Hermitian operator near the boundary. We first review two quick formal arguments — one purely from a bulk perspective, and another assuming AdS/CFT. Then we describe this construction in more detail and explain how at large $N$, the low-energy Fock space can be generated explicitly in the form above.

**Formal arguments**

From a bulk perspective, we have a number of weakly interacting fields at low-energies. These fields include the low-energy gravitons but may also include stringy excitations. At low-energies, it is possible to frame all dynamics with the Hilbert space formed by

$$\mathcal{H} = \text{span of}\{X(\tau)|0\rangle\},\tag{54}$$

where the $X(\tau)$ are simple Hermitian operators at an arbitrary point of time i.e. $\tau$ can range from $(-\infty, \infty)$. This is true since time-evolution manifestly evolves this set back to itself. But then standard analyticity arguments (see Appendix A of [1]) tell us that Hermitian operators from a smaller time-band $[0, \epsilon]$ generate a space of states that is dense in the Hilbert space above.

A second argument, which assumes AdS/CFT, is as follows. The set of bulk operators near the boundary are dual to boundary operators by the extrapolate dictionary [50]. Now, it is clear, from the state-operator map in the conformal field theory that the set of states obtained by applying boundary Hermitian operators on a time-slice to the vacuum form a complete basis for the Hilbert space. If we restrict to simple Hermitian operators — boundary generalized free-fields and low-order polynomials in these generalized free-fields — we obtain the low-energy Hilbert space in the bulk. We use a time-band in this paper, rather than a time-slice, which removes the need for using operators with time-derivatives in the state-operator map.

**Explicit constructions**

We now demonstrate how this construction works explicitly in low-energy bulk effective field theory. Consider a weakly interacting bulk field of dimension $\Delta$, which we denoted by $\phi$ in the text, with boundary value $O$. In global AdS, this operator can be expanded as

$$O(t, \Omega) = \sum_{n,\ell} \sqrt{G_{n,\ell}}\, a_{n,\ell} e^{-i(\Delta + 2n + \ell)t} Y_\ell(\Omega) + \text{h.c},\tag{55}$$

where the operators obey $[a_{n,\ell}, a_{n',\ell'}^\dagger] = \delta_{nn'}\delta_{\ell\ell'}$ and the function $G_{n,\ell}$ is explicitly given by

$$G_{n,\ell} = \frac{4\pi^d \Gamma(\Delta + n + \ell)\Gamma(\Delta + n + 1 - d/2)}{\Gamma(d/2)\Gamma(n+1)\Gamma(\Delta)\Gamma(\Delta + 1 - d/2)\Gamma(d/2 + n + \ell)}.\tag{56}$$

The normalized low-energy single-particle states are simply created by

$$|n, \ell\rangle = a_{n,\ell}^\dagger |0\rangle.\tag{57}$$

Now, it is obvious that one way to create such states through a Hermitian operator localized near the boundary is to consider

$$|n, \ell\rangle = \frac{1}{\sqrt{G_{n,\ell}}} \int_0^\pi O(t, \Omega) e^{-i(\Delta + 2n + \ell)t} Y_\ell^*(\Omega) d^{d-1}\Omega \frac{dt}{\pi}|0\rangle,\tag{58}$$

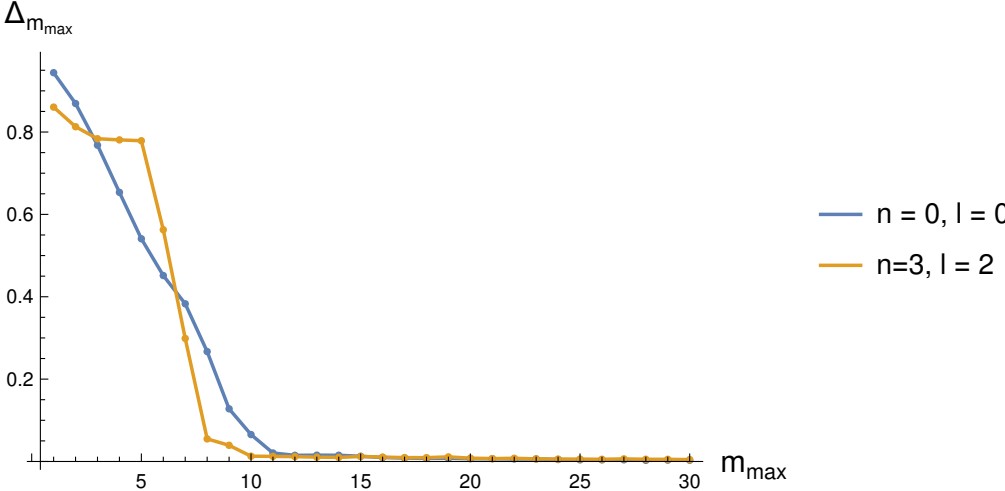

Figure 4: *The figure displays the error $\Delta_{m_{max}}$ in approximating a state as a function of $m_{max}$ for two low-energy states. It is evident that this declines monotonically as $m_{max}$ is increased and becomes very small.*

but this is *not* what our observers need since it involves an integral over the light-crossing time of AdS. The claim above is that we *can also generate* the state by smearing the bulk field in the time-band $[0, \epsilon]$.

A simple algorithm to numerically approximate the state is as follows. Consider the window function

$$w(t) = \theta(t)\theta(\epsilon - t)e^{\frac{1}{t(t-\epsilon)}}, \tag{59}$$

which is a real function that vanishes smoothly at the end-points of $[0, \epsilon]$. Now, consider the states

$$|h_m\rangle = \int_0^\epsilon \left[ O(t, \Omega)Y_\ell^*(\Omega)e^{imt}w(t)d^{d-1}\Omega dt \right]|0\rangle = \sum_{n=0}^\infty h_{n,m}\sqrt{G_{n,\ell}}|n, \ell\rangle, \tag{60}$$

where

$$h_{n,m} = \int_0^\epsilon e^{i(\Delta + 2n + m + \ell)t}w(t)dt. \tag{61}$$

We take states from the basis above with values of $m$ ranging between $-m_{\max}$ to $m_{\max}$ and look for coefficients, $c_m$, that minimize

$$\Delta_{m_{\max}} = \left| |n, \ell\rangle - \sum_{m=-m_{\max}}^{m_{\max}} c_m|h_m\rangle \right|^2.$$

The arguments above tell us that this error term converges to 0 as the value of $m_{\max}$ is increased. The original state is thus approximated better and better by a sum of states dual to operators in a small time band.

An extremely simple Mathematica code to generate a numerical approximation to the state $|n, \ell\rangle$ can be found at [51] . Note that, in this code, apart from putting a cutoff on $m_{\max}$, it is also necessary to truncate the sum over $n$ in (60) to some finite value $N_{\text{cut}}$. Figure 4 shows a plot of $\Delta_{m_{\max}}$ vs $m_{\max}$ for two possible choices of $n, \ell$ for a massless bulk field in $d = 4$, with $N_{\text{cut}} = 100$. As is evident, this error term consistently tends to 0.

Once we have constructed single-particle states, multi-particle states are easy to construct. In the weakly coupled limit, the structure of the Hilbert space is that of a Fock space. So multi-particle states can be constructed just by multiplying operators of the form above, and then acting on the vacuum after smearing them appropriately.

The careful reader may notice a subtlety. In the analysis above, we have used the back-reaction of bulk excitations on the metric, which appears only during interactions, and yet we are content with using the Fock space approximation here. The reason is that any effect of interactions is only at *subleading order* in $\frac{1}{N}$, and therefore it does not affect the observers ability to distinguish the bulk state at leading order at all. In particular, say that the observers wish to find the operator dual to a state $|X\rangle$ and they use an approximation so that

$$X_{\text{approx}}|0\rangle = |X\rangle + \frac{1}{N}|X_{\text{err}}\rangle. \tag{62}$$

The reader can check by examining the procedure described in section 3 that this error makes only a $\frac{1}{N}$ difference in all measurements conducted by the observer. Since the accuracy of the task set for the observers is set by $\delta \gg \frac{1}{N}$ this error is unimportant.

For this reason, one can also use the *free* boundary-bulk smearing function given in [2]; map the boundary operators above to bulk fields on the slice $t = \frac{\epsilon}{2}$, and with radial coordinate $r \in [\cot\frac{\epsilon}{2}, \infty)$ and use these smeared bulk fields to generate the state.

In this paper, for simplicity, we have considered the case where all interactions are controlled by $\frac{1}{N}$. But, our protocol would work even if there was a hierarchy of interactions. If interactions between quantum fields were controlled by a parameter $\lambda \gg \frac{1}{N}$ but with $\lambda \ll 1$, then we would need to correct the operators above perturbatively in $\lambda$. This calculation can still be done reliably within the framework of QFT in curved spacetime.

## B  Orthogonalizing unitaries

In section 3, we explained that if the observers encountered a state with $|\langle 0|g\rangle|^2 \neq 0$, they could act with a preliminary unitary $\mathfrak{U}^z$ with the property that $\langle 0|\mathfrak{U}^z|g\rangle = 0$. They could then apply the protocol of section 3 to $\mathfrak{U}^z|g\rangle$, and back-calculate $|g\rangle$. We now explain how to construct the unitary $\mathfrak{U}^z$.

Although, we present the construction slightly formally below, the basic idea behind the construction of this unitary is quite simple. Using a single propagating field, and by smearing the field and the conjugate momentum over a small region of spacetime, the observers can construct one simple-harmonic degree of freedom. The unitary they need to find acts only on this simple-harmonic degree of freedom. By rotating the states of this degree of freedom appropriately, it is always possible to make the state $\mathfrak{U}^z|g\rangle$ orthogonal to $|0\rangle$.

Let $O(t, \Omega)$ be the boundary value of a bulk field as defined in Appendix A. Then by a suitable choice of two functions it is possible to define operators

$$\mathcal{X} = \int dt d^{d-1}\Omega O(t, \Omega) f_1(t, \Omega); \qquad \mathcal{P} = \int dt d^{d-1}\Omega O(t, \Omega) f_2(t, \Omega), \tag{63}$$

so that these operators obey the Heisenberg algebra

$$[\mathcal{X}, \mathcal{P}] = i. \tag{64}$$

Here the functions $f_1, f_2$ have limited support in the time band $t \in [0, \epsilon]$ and also on the boundary sphere. There is no unique choice of $f_1$ and $f_2$ and so we do not write down their forms explicitly. The relation above is valid in the large-$N$ approximation where the commutator of a bulk field with itself is well-approximated by a $c$-number. By the same reasoning as in Appendix A, it is acceptable to use this approximation while constructing the orthogonalizing unitary.

We can also define the operators

$$\mathcal{A} = \frac{1}{\sqrt{2}}(\mathcal{X} + i\mathcal{P}); \qquad \mathcal{A}^\dagger = \frac{1}{\sqrt{2}}(\mathcal{X} - i\mathcal{P}). \tag{65}$$

The Hilbert space of the full theory furnishes a representation of the algebra corresponding to this simple harmonic degrees of freedom i.e. the algebra that can be expanded in terms of the $\mathcal{X}$ and $\mathcal{P}$ operators. One of the useful elements of this algebra is the projector onto the zero-eigenspace of the number operator

$$\mathcal{P}_0^{\text{sho}} = \int_0^{2\pi} e^{i\theta(\mathcal{A}^\dagger \mathcal{A})} \frac{d\theta}{2\pi}. \tag{66}$$

Note that this projector projects onto an infinite dimensional subspace in the full theory. Using this projector we can also construct projectors onto other number eigenspaces.

$$\mathcal{P}_n^{\text{sho}} = \frac{1}{n!}(\mathcal{A}^\dagger)^n \int_0^{2\pi} e^{i\theta(\mathcal{A}^\dagger \mathcal{A})} \frac{d\theta}{2\pi} (\mathcal{A})^n. \tag{67}$$

These operators furnish a partition of the identity so that $\sum \mathcal{P}_n^{\text{sho}} = 1$. One can also construct more general "transition operators"

$$\mathcal{T}_{m,n}^{\text{sho}} = \frac{1}{\sqrt{m!n!}}(\mathcal{A}^\dagger)^m \int_0^{2\pi} e^{i\theta(\mathcal{A}^\dagger \mathcal{A})} \frac{d\theta}{2\pi} (\mathcal{A})^n. \tag{68}$$

Note that $\mathcal{P}_n^{\text{sho}} = \mathcal{T}_{n,n}^{\text{sho}}$, and also that the operators $\mathcal{T}_{m,n}^{\text{sho}}$ furnish a complete basis for all elements of the algebra.

Now consider a unitary $\mathfrak{U}^z$ that is within this algebra. Since the transition operators form a basis, we can expand

$$\mathfrak{U}^z = \sum_{n,m=0}^{\infty} u_{nm} \mathcal{T}_{n,m}^{\text{sho}}, \tag{69}$$

where $u_{nm}$ is just a matrix of $c$-numbers. The condition that $\mathfrak{U}^z$ is unitary is the same as the condition that the $c$-number matrix $u_{nm}$ is unitary. So we have

$$\langle 0|\mathfrak{U}^z|g\rangle = \sum_{n,m=0}^{\infty} u_{mn} \langle 0|\mathcal{T}_{n,m}^{\text{sho}}|g\rangle. \tag{70}$$

Once again $t_{nm} = \langle 0|\mathcal{T}_{n,m}^{\text{sho}}|g\rangle$ is just some matrix of $c$-numbers. So the observers need to find a unitary matrix that satisfies a single linear constraint.

$$\sum_{n,m=0}^{\infty} u_{mn} t_{nm} = 0. \tag{71}$$

The problem (71) is a linear-algebra problem of finding a unitary matrix with the property that when multiplied with another matrix and traced, it yields zero. It is not difficult to see that such a unitary matrix can always be found for any choice of $t_{nm}$. But for the sake of completeness, we now give a proof.

We will give a proof in steps. First consider the two-dimensional version of this problem, where we are given an arbitrary $2 \times 2$ matrix $t_{nm}$ and need to find a $2 \times 2$ unitary matrix, $u_{nm}$, so that $\text{Tr}(ut) = 0$. We express the unitary matrix in terms of a unit-vector on the sphere, $\hat{n}$, and an angle $\theta$ as $u_{nm} = \cos(\frac{\theta}{2})\delta_{nm} + i(\vec{\sigma} \cdot \hat{n})_{nm} \sin(\frac{\theta}{2})$. Then we first

choose $\hat{n}$ to be orthogonal to the vector $\text{Re}\left(\text{Tr}(\vec{\sigma}\,t)\right)$. Then we are left with the single equation $\text{Tr}(t)\cos(\frac{\theta}{2}) - \text{Im}\left(\text{Tr}((\vec{\sigma}\cdot\hat{n})\,t)\right)\sin(\frac{\theta}{2})$ which can be solved by choosing $\theta$ appropriately.

Now, consider the $D$-dimensional problem. If we collect the columns of $t$ into a set of vectors $\vec{t}_1\ldots\vec{t}_D$ we need to find a set of orthonormal basis vectors $\vec{v}_1,\ldots\vec{v}_D$ that satisfy $\sum\langle\vec{v}_i,\vec{t}_i\rangle = 0$, with the usual conjugate-bilinear inner-product. The unitary matrix is then obtained by taking $\vec{v}_i$ to be the columns of $u^{\dagger}$.

Such an orthonormal basis can be found as follows. We choose $\vec{v}_D$ to be orthogonal to $\vec{t}_D$. Then we choose $\vec{v}_{D-1}$ to be orthogonal to $\vec{v}_D$ and $\vec{t}_{D-1}$. We similarly choose $\vec{v}_P$ to be orthogonal to $\vec{t}_P$ and $\vec{v}_{P+1},\vec{v}_{P+2}\ldots\vec{v}_D$ for all vectors until $P = 3$. Then $\vec{v}_1$ and $\vec{v}_2$ are confined to a two-dimensional space, since they must be orthogonal to $\vec{v}_3\ldots\vec{v}_D$, and further they must satisfy $\langle\vec{v}_1,\vec{t}_1\rangle + \langle\vec{v}_2,\vec{t}_2\rangle = 0$. But this is just the two-dimensional problem that we solved above.

We now turn to the infinite dimensional case (71). On physical grounds, we expect that $\langle 0|\mathcal{T}_{n,m}^{\text{sho}}|g\rangle \ll 1$ when $n,m$ are much larger than the "occupancy" in the two states i.e. when $n,m \gg \langle 0|\mathcal{A}^{\dagger}\mathcal{A}|0\rangle$ and $n,m \gg \langle g|\mathcal{A}^{\dagger}\mathcal{A}|g\rangle$. So we consider a unitary that does not act on states with occupancy beyond some $D$ i.e. $u_{ij} = \delta_{ij}$ for $i,j \geq D$ and $u_{ij} = 0$ if $i \geq D, j < D$ or $i < D, j \geq D$. Then we just need to deform the finite dimensional solution above so that $\sum_{n,m=0}^{D-1} u_{mn}t_{nm} = -\sum_{i\geq D} t_{ii}$. This can clearly be done provided the right hand side is small enough, which can be achieved by taking $D$ to be large enough.

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
