# Peer review of "A physical protocol for observers near the boundary to obtain bulk information in quantum gravity"

_SciPost Physics, doi:SciPost Phys. 10, 106 (2021)_

## Round 1 · Referee Report · Samir Mathur (Referee 1) · 2020-10-17

Strengths

This paper is clearly written; this is very useful for a topic which relates to the area of information and black holes, as there are many diverse opinions in this area of work.

Weaknesses

I do not find myself in agreement with the main physics ideas presented in the paper.

Report

This is a clearly written paper, but I do not find myself in agreement with the main claims. It is possible that I have not understood the arguments, so I will try to give a detailed explanation of what I think are the problems with the general approach of this and similar papers.

The paper argues that theories with gravity are special in that they allow a determination of low energy states of a system from measurements at the boundary. The process of determination involves some extensions of the idea that if all states are assumed to have different energies, then measuring this energy from infinity will identify the state. I list my difficulties below:

(1) Suppose I have a particle in a box, but I do not know if it is a proton or an electron. From measurements far away, I can measure the small electric field produced by the particle, and find out which particle is in the box. This is not surprising; it is just a consequence of having a long range potential sourced by the charged particles.

The way gravity is used in the paper does not look fundamentally different from this, though the authors argue that gravity is indeed different from gauge theories. It is true that with the electric field one cannot identify neutral excitations. Is it being argued that gravity is special because it couple to everything? (Also, in AdS3 the gravity theory can be recast as a Chern-Simmons gauge theory, and more generally gravity can be written as a gauge theory of local frame rotations.)

(2) In their example on pg 3, the authors start with a state |g> peaked near the center of AdS, act on this with an operator X near the boundary and ask for the amplitude to get the vacuum |0>. They write the amplitude for this as <X|g>, and ask of this is close to <X|X>; if it is, then one has shown that the state in the interior is |g>.

I would have expected <X|g> to be very small (i.e. not equal to <X|X> ~1), since X operates in a region where |g> has a small tail. What is the physical picture that the authors have in mind here? Are they arguing for nonlocal effects here? For example, suppose we are allowed to make local unitary operations on a sphere with a large radius. Are they arguing that such operations can instantaneously create an object at the center of the sphere? (They are doing all their operations close to a time t=0, so there would not be time for signals to travel from the sphere to its center. ) Is it being argued that causality as we normally understand it is violated in a large way when the theory contains gravity?

(3) The authors are focused on whether they can identify a state localized near the center by measurements at the boundary. While they do not focus on issues connected to black hole physics, they do mention that the questions they are looking at related to quantum information questions in black holes. With the traditional picture of the hole (i.e., when the hole has a horizon) we can have both positive and negative energy excitations in the region inside the horizon. In this situation one can make an arbitrarily large number of states that are arbitrarily close to each other in the energy that is observed at infinity. Thus I do not understand how measuring the energy at infinity can help with the problem of black holes. (With the fuzzball paradigm there is no region that is inside a horizon, so there are no negative energy modes; but in this paradigm the black hole microstate is just like a piece of coal, so there is no information paradox to think about.)

(4) The authors are concerned with trying to identify the state |g> near the center, from measurements far away. But such a measurement is not pertinent to the puzzle associated with black holes. This puzzle is termed the 'information paradox', but this is a misnomer; what we have is actually an 'entanglement problem' between the radiation quanta and the hole.

To understand this point in the context of the present paper, consider an electron at the center and another at the boundary, entangled in a singlet state 01-10. We can certainly know the state of the electron in the interior if we measure the state of the electron at the boundary. But this does not mean that there is no entanglement between the two electrons!

Further, In line with what this paper proposes, we can measure the spin of the electron at the center by looking at the small frame dragging effects on the metric near the boundary. Again, this does not mean that there is no entanglement between the two electrons. The black hole puzzle is that if the electron at the center disappears from the universe (when the black hole in which it is contained evaporates) then the remaining electron at the boundary has no well defined state. This problem has nothing to do with 'knowing' the state of the electron at the center. The entanglement of the electron at the boundary implies that if we pass this electron through a Stern Gerlach placed at any angle, there will be an equal probability for the electron to show up in either branch of the apparatus. By contrast, if the electron at the boundary was in a non-entangled state like 0+1, then a Stern-Gerlach placed along the x-axis would see the electron always pass through the +x branch of the apparatus. Thus entanglement is a simple measurable property of an electron, and has nothing to do with 'knowing' the state of the 'other' electron. The authors talk about the uncertainty related to entanglement in the standard language of Von Neumann entropy. But this uncertainty or lack of information is not an uncertainty about the state of the other electron (the one in the interior). Rather, it is the uncertainty of the boundary electron itself in the sense that we will never be able to say which branch of the Stern-Gerlach it will pass through.

(5) The authors base their arguments on a few computations; let me list my difficulties with these computations:

(i) In eq. 2.5 they expand the unitary operator to first order as 1+iJX. I am worried about not keeping the second order term here, since it is these second order terms that help give causality in field theory. Take a free scalar field \phi in 1+1 dimensions. Apply a unitary O_1=e^{i JX} at (t,x)=(0,0), and O_2=e^{-iJX} at (t,x)=(\epsilon, D). Thus the two points of application are spacelike separated. Now compute <0|T[O_2 O_1 ]|0>. This gives J^2<0\phi(\epsilon,D)\phi(0,0)|0> which is nonvanishing. It would therefore appear that a unitary operation at the first point can be detected at the second point, which is of course in conflict with causality. To resolve this conflict one has to expand the exponentials to one more order. The authors should check that a similar effect is not leading to apparently acausal effects in their computations.

(ii) On pg 8 the authors say that energy gaps in AdS are of order the inverse AdS radius. But consider AdS3 X S^3. Consider a graviton with a large angular momentum j\gg 1 on the sphere. The radial energy levels in the AdS for this graviton are quantized in units of 1/j times the inverse AdS3 radius. Thus the authors should check carefully what energy gaps are allowed for massive particles in AdS, where the mass can come from rotation on the sphere or because we considered a heavy string state.

(iii) The authors give a computation in their Appendix A which seems to say that knowing a field in a small region near the boundary can allow a determination of its value in the interior. I am concerned about this, in view of the following toy example. Consider functions on the line interval (-1, 1). Let f1=1 be the constant function on this interval. Let f2=1 for x in the range (0,1), and f2=1-Exp[1/x] for x in the range [-1,0]. Thus f1 and f2 are the same for x>1 but differ in a completely smooth (C_\infty) way in the region x<1. If we are given only access to the range of x given by (1-\epsilon, 1), then we will not be able to distinguish these functions. Why is the situation different for the case with Hilbert space states that the authors consider? If I take a fermion field, then can I not make exactly the above two wavefunctions with local operators, and therefore be unable to distinguish the states from the boundary?

(7) In summary, I am not in agreement with the general idea of the paper that there are interesting effects at the level of perturbative gravity that imply a revision of our thinking about locality and causality. Of course it is possible that I have not understood some of the arguments, and if this is the case, I apologize in advance for being critical of them.

  • validity: low
  • significance: ok
  • originality: high
  • clarity: top
  • formatting: perfect
  • grammar: perfect

Author:  Suvrat Raju  on 2020-10-20  [id 1012]

(in reply to Report 1 by Samir Mathur on 2020-10-17)

We are grateful to Prof. Mathur for his detailed report, and also for signing the report, which helped us understand the referee's perspective.

It is true that our paper reaches a novel and striking conclusion. So we are delighted to have the opportunity to answer Prof. Mathur's questions since other readers may have similar questions.

We believe that the main strength of this paper is that its conclusions are based on a simple and unimpeachable technical calculation. So we would like to start by addressing point (5) in Prof. Mathur's report, which is the only point where he mentions any technical objections. The other points are more general, and so we will answer them later.

(5)
(i) Prof. Mathur suggests that we are "not keeping the second order term here." But this is incorrect. In fact, we are very careful about this second order term in our analysis. It is displayed explicitly in 1.1, and the reason it is not displayed in eqn. (2.5) is mentioned immediately below that equation and again below eqn. (3.1): the reason is that this term drops out of the analysis.

Let us reiterate the reason that this term drops out. The unitary, U, always acts on states $|g \rangle$ that satisfy $\langle 0 | g \rangle = 0$. We are always interested in measuring $P_0$ after acting with the unitary. If we take the unitary to coincide with $e^{i J X}$ to second order in $J$, its action is $U |g \rangle = |g \rangle + i J X |g \rangle - (J^2 X^2/2) |g \rangle$. Now consider what happens when we sandwich $P_0=|0 \rangle\langle 0|$ between the state and its dual bra. The answer is clearly $\langle g|U^*|0 \rangle\langle 0|U|g \rangle = |\langle 0|U|g \rangle|^2$. Now $\langle 0|U|g \rangle = i J \langle X|g \rangle - J^2 \langle 0|X^2|g \rangle/2$ where the leading "1" in the unitary goes away because $\langle 0|g \rangle = 0$. When we now consider $|\langle 0|U|g \rangle|^2$ we see that the second order term in $J$ only contributes to $J^3$ and higher order terms!

Note that, for this second order term to drop out, it was important that $\langle 0|g \rangle = 0$. This is the reason we act with the unitary $\mathfrak{U}^z$ in step 1 of our protocol (Figure 2) and this unitary is the subject of Appendix B.

So far from ignoring the second order term, we believe that we have devoted significant attention to ensure that it does not enter the final answer. We can emphasize this even more in our revised paper.

The example that Prof. Mathur gives where one expands both exponentials to first order and then reaches an incorrect result when the exponentials are multiplied (because both exponentials also contain a factor of 1), is not what is happening in our calculation. There is no analogue of this effect in local field theories.

(ii) In AdS/CFT, the energy levels in global AdS that we are referring to are dual to the spectrum of operators primaries in the CFT. Prof. Mathur is right that the spectrum of operator primaries is not integrally quantized but our assumption is only that the spectrum of primaries is discrete and the number of primaries below a fixed cutoff is finite. We believe that this is a very reasonable assumption, but we would be glad to mention it explicitly.

(iii) We believe that Prof. Mathur has misunderstood the point of Appendix A. We certainly do not claim that knowing any function in a small interval allows one to reconstruct it in a larger interval, which would be obviously incorrect as Prof. Mathur points out.

The argument of appendix A is that any state created by smearing an operator with a function in the time-band $[0, \pi]$ can be reproduced by smearing the same operator with a different function in the time-band $[0, \epsilon]$.

We have already referred to the appropriate literature in the text but for convenience, we reiterate the argument here. Consider the state $\int_0^{\pi} O(t) f(t) | 0 \rangle$ . Expanding both O and f in Fourier modes, this state is just $\sum_{n > 0} O_{n}^{\dagger} f_{n} | 0 \rangle$. It is important that the sum above runs only over positive $n$ since the energy of the vacuum cannot be lowered, and so all the negative frequency parts of $f$ drop out. Therefore if we want to find another function $g(t)$ with support in $[0, \epsilon]$ so that $\int_0^{\epsilon} O(t) g(t) |0 \rangle = \int_0^{\pi} O(t) f(t) |0 \rangle$, we only need to make sure that the positive frequency part of $g$ coincides with the positive frequency part of $f$.

Given a function with support in a larger interval, one can always find a function with support in a smaller interval, so that the two functions agree arbitrarily well in their *positive-frequency parts*. This can be proved using elementary complex analysis. In the Appendix, we have also verified this result numerically, and our code is publicly available.

We would like to emphasize, once more, that the argument of Appendix A holds even in local QFTs and so, by itself, it does not tell us that we can completely identify the state from a small time band. It only tells us that we can produce all states by acting with operators from a small time band. (Note: "produce state" $\neq$ "identify state"). To be able to identify states, we need to combine this argument with the projector on the vacuum, which is the subject of the main text of the paper.

We believe that the above explanations completely address the technical points raised by Prof. Mathur. We hope that Prof. Mathur will agree that our technical arguments are correct, and if he has any doubt about the correctness of any equation in the paper, we would be glad to address it.

We now turn to the other questions in the referee report, which are of a more general nature.
(1) Gravity is very different from gauge theories because the projector on the vacuum selects a unique state. In gauge theories, one can project onto states of zero charge but there are an infinite number of such states. This is why one can obtain information about the bulk from the boundary, whereas one cannot do so in non-gravitational gauge theories.

This is a physical fact that utilizes the positivity of energy, which does not change if one formulates the theory differently.

In Prof. Mathur's language this is indeed because there are no localized neutral excitations in gravity, whereas there are such excitations in gauge theories.

(2) $X$ is indeed an operator near the boundary but, as we explained above, $X |0 \rangle$ can create an arbitrary state including the state $|g \rangle$. This does not involve any nonlocality and is true in an LQFT.

The magic in gravity is that the matrix element $\langle 0|X|g \rangle$ becomes physically accessible. This matrix element is not accessible to a near-boundary observer in any theory without gravity.

Points (3) and (4) are both about black holes. The focus in this paper is not on black holes and --- even though we recognize that Prof. Mathur may not agree with our perspective on black holes --- we would like to keep this discussion brief in order to keep the discussion on track. We answer (4) before (3).

(4) In the example of an EPR pair, that Prof. Mathur gives, it is possible to act with a unitary operator on the electron in the center. In a non-gravitational theory, the action of this unitary is completely invisible to the observer near the boundary. Since the near-boundary observer has no way of detecting whether or not the observer in the center has acted with this unitary , in the language of this paper, we would say that the observer near the boundary cannot obtain "information" about the electron in the center.

In this paper we have shown that, in the presence of gravitational effects, the observer near the boundary can determine whether or not the unitary was applied in the center. There is no analogue of this in qubit systems with factorized Hilbert spaces.

In the context of black holes, this means that a copy of the information in the middle of the black hole interior remains in the exterior. When the black hole evaporates away, this copy still remains and this is why there is no information loss.

We emphasize that our arguments involve no ad-hoc postulates about nonlocality and follow from a careful semiclassical analysis.

(3) We do not claim anywhere that a measurement of the energy is sufficient to completely distinguish between states. This cannot be the case since, clearly, there can be degeneracies in energy-levels even at low energies. Moreover, energy measurements, by themselves, cannot determine the relative phase in a linear combination of energy eigenstates. The claim is that measurements of the energy and also of correlators of the energy with other observables suffice to determine the state. In this paper, we have shown this explicitly at low energies. The formal arguments that we allude to imply that this continues to be true for high-energy states like black holes.

We hope that this comprehensively addresses Prof. Mathur's concerns and that the paper can proceed towards publication.

---

## Round 1 · Referee Report · Anonymous (Referee 2) · 2021-1-19

Strengths

The paper states clearly what it tries to achieve. The proposed answer is appealingly simple.

Weaknesses

A strong claim is made on the basis of a (perhaps) deceptively simple proposal. The devil may hide in the details.

Report

This paper suggests a protocol to determine the quantum state of a gravitational system using only measurements in a small boundary band of AdS (or alternatively a small annular region near the boundary of a Cauchy slice that runs through the bulk).

The paper is thought provoking and appealing in its simplicity. The basic algorithm to determine the boundary state has a similar flavour as basic quantum-informational protocols.

On the other hand its simplicity is perhaps deceptive. Before recommending publication I would like the authors to clarify a few points and comments.

1. Practicality of their protocol a): a typical observer (inasmuch as an observer in AdS can be called typical) only has access to a single AdS space. I am not convinced by the authors’ suggestion of using a super observer who instead lives in a spacetime with many, presumably a very large or even infinite number of, identical copies. For the protocol to be useful, it should be practically be implementable in a single copy of the Universe they live in.

2. The protocol involves measuring the local value of the metric. This seems to be a gauge-variant quantity. Could the authors be clearer about what exact measurements the observer needs to make?

3. Practicality of the protocol b): as the authors mention, in practice the fraction of measuring zero energy can only be exactly determined by performing an infinite number of experiments. This means the protocol naturally involves a tolerance parameter $\delta$. Throughout the paper we meet various other notions of cutoffs (e.g. to define low-energy), semi-classical parameters (1/N) and so on. It would be highly useful if the authors could clearly state at some point in their manuscript how all the relevant scales interact, and where exactly the allowed window for the protocol is to be found. Does this effectively lead to a bound on the allowed non-locality the protocol seems to suggest?

4. The protocol relies heavily on the energy being quantised. Are the authors saying that their protocol would not work for example in the Poincaré path of AdS? If so, how are we to think of this? Furthermore this seems to pose a strong possible obstruction of implementing a similar protocol for more general spacetimes, where energy may very well not be quantised.

  • validity: good
  • significance: high
  • originality: high
  • clarity: high
  • formatting: perfect
  • grammar: perfect

Author:  Suvrat Raju  on 2021-01-29  [id 1188]

(in reply to Report 2 on 2021-01-19)

We would like to thank the referee for the helpful questions and comments.

Below, we have answered the referee's questions and also provided details of the changes made in response to the referee's comments.

1) Many copies

The necessity of many copies has no specific relation to our protocol but arises from the inherent features of quantum mechanics. Since the results of observations are probabilistic, in general, it requires multiple observations on identically prepared systems to obtain quantum information. This is true, not only for quantities like the entanglement entropy, but even for something as mundane as bulk correlation functions. Since correlators are expectation values of products of operators, to measure a correlator requires the observers to prepare many identical copies of the system, make measurements in each copy, and then average the results.

Therefore, even if the observers were to explore the bulk to obtain information, and not use our protocol at all, they would still require multiple identical copies in order to be able to fix the bulk state.

It is possible that by explicitly mentioning this mundane issue, we may may have unintentionally suggested that something unusual is going on. Therefore, we have inserted a remark to emphasize that there is nothing unusual about this aspect of our protocol and it follows from standard quantum mechanics.

2) Measurements of the metric

Thank you for this remark. The energy that we are measuring is the integrated component of $h_{00}$ in the Fefferman-Graham gauge. There is a corresponding gauge-invariant expression, but we wrote the expression in the Fefferman Graham gauge for simplicity. We have inserted a reference and a remark to clarify the issue.

3) Cutoff parameters

Thank you for the remark about cutoff parameters. We introduce two parameters to make the protocol precise. There is a UV-cutoff, $\Lambda$, and an accuracy scale, $\delta$, that can be set independently. But the UV cutoff is much smaller than the Planck scale, so that $\Lambda \ll N$ and the accuracy parameter is larger than the ratio of the cosmological scale to the Planck scale: $\delta \gg \frac{1}{N}$. We have inserted this clarification in the text. These cutoffs are meant to ensure the simplicity of our protocol, and can be relaxed if the observers are allowed to make more complicated measurements as is discussed in Ref. [1] of our paper.

4) Quantization of energy.

From the point of view of the observers near the boundary, there is not much difference between Poincare AdS and global AdS. Since a Cauchy slice for Poincare AdS also provides a Cauchy slice for global AdS, the Hilbert spaces are isomorphic. More specifically, observers near the boundary of ``Poincare AdS'' can always choose to classify states using a linear combination of the Poincare generators (see Luscher and Mack, Commun. math. Phys. 41,203--234 (1975)), obtain a discrete spectrum for these new generators, and then carry out precisely the protocol that we have indicated.

Nevertheless, the referee's question about infrared effects is an important one. Not only is it relevant for flat space, it is also important for extending our protocol to black holes in AdS. In effective-field theory about a black-hole background, the quantization of energy-levels is no longer apparent. This is a very interesting question that we hope to address in forthcoming work. We have added a remark to this effect in the paper mentioning this issue.

In addition to the changes described here, we have also added footnote 1 on the spectrum of operator dimensions. This is in response to a comment in Prof. Mathur's report, as indicated in our reply to that report. We have also added additional remarks to emphasize the fact that we are not ignoring the second-order term but that we do not display it explicitly at all times since it drops out of the analysis.

Finally, we have altered the word "semiclassical'' (which is understood in different ways by different researchers) to "low-energy theory'' so as to be more precise about what we mean.

We hope that this adequately addresses the concerns of both referees and the paper can now proceed towards publication.

---

## Round 2 · Referee Report · Anonymous (Referee 2) · 2021-3-21

Strengths

see my first report

Weaknesses

see my first report

Report

I would like to thank the authors for replying to my comments and questions and for introducing modifications to their manuscript where necessary. I recommend publishing the paper in the present form.

---

## Round 2 · Referee Report · Samir Mathur (Referee 1) · 2021-4-8

Report

I remain puzzled about the results of this paper. However I have had a chance to understand the arguments better, so perhaps I can focus my concerns better as well. Notwithstanding these concerns, I think this paper should be published, since the authors have spent time thinking about these issues, and it would be useful to have the arguments available to the community to spur debate on these interesting issues.

From what I understand, the authors claim that in any theory with gravity, one can find the state inside some region $r < R$ by doing measurements only in a region $r>>R$; thus in particular light has not had time to bring any signals from the inner region to where the measurements are being done. Gravity can remain weak throughout, so no novel features of string theory or nonperturbative quantum gravity are being used.

The essence of the argument seems to be the following. The gravitational field h couples to the matter energy E. By doing careful measurements of the weak gravitational field near infinity, we can project the overall state to an eigenstate of the energy E. But energy eigenstates are typically travelling waves, which will keep oscillating all the way to infinity (rather than being localized wavepackets). Then we can observe this state near infinity, and thus infer what it must have been in the interior.

In particular, let the theory have two identical scale fields $\phi_1, \phi_2$. We make a localized wavepacket of one of these fields in the region $r<R$. We wish to detect from infinity which of the two fields we excited. We measure h at infinity, projecting to an energy eigenstate E for the scalar fields; then looking at the corresponding travelling wave of $\phi_i$ that reaches infinity in such an eigenstate, we will know which scalar field was excited in the interior.

One can worry about such an argument because of the simpler situation with just a scalar field in flat space. The vacuum has correlations at points that are spatial separated; this makes the Feynman propagator nonzero at spacelike separations. But an excitation at a point x cannot be detected at a spacelike - separated point y. To prove this one can do one of two things: (i) Check the vanishing of commutators of operators at these two points; this can be difficult in the gravity theory since one will need a good definition of a local operator first (ii) Put a source at one position and an Unruh -de Wit detector at the other, checking that the latter is not excited; this however is a complicated computation due to cancellations between several terms in the amplitude, and the local operators would have to be made very carefully just as in (i).

So I tried to make a counterexample to their claim as follows. Suppose we can make two solutions of the wavefunctional, satisfying the gauge constraints, with the following property: In the region $r<R$, the two are different, while in the region $r>2R$, they are identical. In this case I believe that no experiment at $r>2R$ can detect the difference between the two solutions at $r<R$. I believe the authors would claim that two such solutions cannot be made, but it was not clear to me that such was the case.

In the attached pdf file I sketch how I would make such a counterexample for the simpler case of the electromagnetic theory; I am guessing that the gravitational case is more messy to construct but similar. Thus my question would be

(i) Would such a counterexample, if it exists, disprove the authors' claim? Or have I completely misunderstood the problem?

(ii) If yes to (i), is there an error in the gauge theory example I have tried to make?

(iii) If the gauge theory counterexample is correct, is the gravity case very different? (The gravity case does have an extra spatial Laplacian term, but it is not clear why this should change the existence of such a construction)

To summarize, I remain confused by the claim of this paper. But as mentioned above, I think the paper should be published. In a complicated field like quantum gravity, it cannot be the authors' responsibility to convince another person (the referee) of their ideas; the correctness of the ideas should emerge by further investigations in the general gravity community.

Attachment

  • validity: -
  • significance: -
  • originality: -
  • clarity: -
  • formatting: -
  • grammar: -

Author:  Suvrat Raju  on 2021-04-16  [id 1367]

(in reply to Report 2 by Samir Mathur on 2021-04-08)

We would like to thank the referee for the latest report and for recommending publication of the paper.

Our response to the second referee report is available in the attached pdf file (p.1 to p. 3). A simple textual summary is also provided below.

Aside from this scientific response, we would like to explain some background that, we believe, is important for readers and editors to understand this exchange.

Background

The question being discussed here is whether gravity localizes quantum information differently from other local quantum field theories.

The referee's perspective on this issue is known to be different from ours. There has been extensive discussion on this issue already in the literature, and some of this difference in perspective can be traced back to alternate resolutions of the black hole information paradox. In arXiv:0909.1038, the referee proposed a "small corrections theorem" that is used to argue that fuzzballs are necessary to resolve the information paradox. In arXiv:1211.6767, it was pointed out that this result tacitly assumes that information is stored locally in gravity, just as it is in local QFTs. Without this assumption, the "small corrections theorem" would fail.

The paper under consideration adds to a growing and already significant body of evidence that gravity does localize quantum information unusually. Some recent independent evidence comes from an analysis of operator algebras in gravity (SciPost Phys. 10, 041 (2021)). But this paper points out that this effect is visible even in the low energy theory of gravity. This paper is not about black holes but the obvious implication of our results is that the qubit models of black hole evaporation used in arXiv:0909.1038 miss an essential aspect of the physics.

Therefore, it is natural that the referee is opposed to the results. To be clear: we have no objection if scientists who are opposed to a program are chosen as referees for a paper that advances the program. This may lead to a productive scientific discourse, as we have had with the referee. But it is very important that, in fairness to the authors, this difference in perspective be accounted for when it comes to making a final judgement on the validity of a paper.

Previous Correspondence

The referee's second report emerges from an extended discussion between the authors and the referee. With the referee's kind permission (p.32 and p.39), we are attaching a transcript of this discussion to this reply. The transcript is available in the attached pdf file, after our response to the referee report. We believe that this correspondence, which covered a number of issues, will be helpful for readers.

The attached transcript runs to over 50 pages, which is considerably longer than our original manuscript.

But we would like to draw the reader's attention to some of the salient features of this transcript. We have repeatedly asked the referee, in our reply to the previous referee report and also in the correspondence (p. 27 and p. 34) to point out any error in any equation or line of text in our paper. No such error has been found or even hinted at.

Second, this correspondence has gone well beyond the traditional confines of peer-review which is usually limited to checking the "correctness" of a paper. Instead, after a few initial questions about the paper itself, the referee has presented us with a sequence of puzzles, drawn from contexts outside of the setup of our paper.

We have been able to resolve all of these puzzles. Thus, it can be seen, in the attached correspondence, that we have resolved puzzles about the difference between gauge and gravity theories, resolved a puzzle about a spin in AdS interacting with a magnetic field and also a puzzle about global symmetries. The most recent discussion has focused on how our picture can be understood from the perspective of wavefunctionals. An explicit error was found in the previous version of the referee's argument. (See p. 37,38) The second referee report attempts to refine this latest argument from the correspondence.

Summary of Response to Report

The referee's "counterexample" is presented in the context of electromagnetism although our paper is about theories of gravity. We are surprised that the referee terms this construction a "counterexample" since the analysis presented is not in contradiction with our results. In fact, we explicitly explain this physical point in our paper. We discuss nongravitational gauge theories in the Introduction and devote an entire subsection, subsection 4.2, to explaining the difference between such theories and gravity. Here we explain that electromagnetism has a a "split property" just like local quantum field theories. Therefore, in such theories, information about a state is not available at infinity. (The only special case, which we describe in section 4.2.1, is where strong priors are given to the observers.) This was also explained in the attached correspondence (p. 8 and p.25 and 26) and the referee notes this explicitly in the referee's own note in the correspondence (p. 41). In fact, the referee's construction in the report aligns with a construction that we ourselves outlined (p. 37).

Perhaps the referee terms this a "counterexample" because of the claim towards the end of the report that this construction can be generalized to gravity. But no justification is given for this claim, and no explicit construction is provided either. Instead the referee only states ''It is not clear to me that a construction similar to the electromagnetic case will not work.''

It is rather simple to see that such a generalization is impossible.

The difference between gravity and electromagnetism is the following. In the electromagnetic case, once we specify that the integral of the electric field at infinity is $Q$, the integral of the bulk current density, $\rho(x)$, is constrained to be $Q$. However, this is not a significant constraint since the current density at different points commutes: $[\rho(x), \rho(x')] = 0$. So $\rho(x)$ can be chosen independently at each point in the bulk and there are an infinite number of configurations of $\rho(x)$ that lead to the same charge as seen at infinity.

In gravity, the integral of the metric over the sphere at infinity again constrains the integral of the bulk energy density $H_{\rm{bulk}}(x)$. But this is now a significant constraint. For instance, in global AdS, if the metric at infinity indicates that the energy is $0$ then there is a unique bulk configuration that is consistent with this constraint, and there is no freedom left in the choice of the bulk wavefunction. Similarly, if the metric at infinity indicates that the energy is $E$, in AdS units, there are a finite number of bulk configurations that are consistent with this constraint since there are a finite number of bulk states with energy $E$.

One might wonder why one cannot repeat the trick used in electromagnetism by choosing $H_{\rm bulk}(x)$ independently at each bulk point while only keeping its integral constant. The difference is that the commutator $[H_{\rm bulk}(x), H_{\rm bulk}(x')]$ leads to the so-called ''Schwinger terms'' that are proportional to $\delta'(x - x')$. In a lattice regularization, this would mean that the energy density at one lattice point does not commute with the energy density at an adjacent lattice point and so the energy density operators at different points in space cannot be diagonalized simultaneously. And this, of course, is why the ground state of the system cannot be found by independently minimizing the energy at each lattice site.

So an attempt to generalize this construction of a split state from electromagnetism to gravity fails since the specification of the metric at infinity does not allow an arbitrary prescription of $H_{\rm bulk}(x)$ in the bulk.

As mentioned above, we have explained this detail in the beginning of the attached pdf file in the note titled ''Author response to the second referee report.''

Summary

To summarize: our paper has undergone intensive scrutiny. We have successfully addressed a large number of questions and puzzles, including the latest "counterexample" presented by the referee even though these discussions have taken us well beyond the original scope of the paper. We understand that the referee may still have reservations about the results. But we would like to respectfully point out --- and the reader can easily verify this by perusing the correspondence above --- that not a single error has been found in our manuscript in this entire process.

We believe that this is a strong indicator of the correctness and quality of the manuscript.

Attachment:

correspondence_with_notes.pdf

---

## Round 2 · List of Changes

The list of changes is also provided in the reply to referee 2 and the changes are explained there. 1) The word "semiclassical" has been changed to "low-energy" at various points in the text so as to be more precise about what we mean. 2) We have elaborated on the relationship of the various cutoffs in the subsection on "Tasks of the Observers" 3) We have inserted text in the subsection on "Abilities and Limitations.." emphasizing that the need for multiple copies is not special to our protocol and follows from the inherently probabilistic nature of quantum mechanics. 4) We have inserted text in the section on "Allowed manipulations" explaining why the second order term is not displayed. (Its contribution vanishes.) 5) In the section on "Allowed measurements", we have inserted a remark and a reference to a gauge-invariant expression for the energy. 6) We have inserted a footnote in the section on "allowed measurements" explaining why the energy-spectrum is expected to be discrete about the vacuum of global AdS. 7) In the section on "implications for black holes" we have noted that, within effective field theory in a black-hole background, the quantization of energy is no-longer visible because the gaps between energy-levels are too small to seen perturbatively.

---

## Editorial Decision

published